# Antimicrobial Activity of Curcumin in Nanoformulations: A Comprehensive Review

**DOI:** 10.3390/ijms22137130

**Published:** 2021-07-01

**Authors:** Jeffersson Krishan Trigo-Gutierrez, Yuliana Vega-Chacón, Amanda Brandão Soares, Ewerton Garcia de Oliveira Mima

**Affiliations:** Laboratory of Applied Microbiology, Department of Dental Materials and Prosthodontics, School of Dentistry, São Paulo State University (Unesp), Araraquara 14800-000, Brazil; jefftrigo347@hotmail.com (J.K.T.-G.); yuliana.v.chacon@gmail.com (Y.V.-C.); brandaoamanda@yahoo.com (A.B.S.)

**Keywords:** curcumin, drug delivery systems, antimicrobial agents, microbial drug resistance, viruses, bacteria, fungi, photochemotherapy

## Abstract

Curcumin (CUR) is a natural substance extracted from turmeric that has antimicrobial properties. Due to its ability to absorb light in the blue spectrum, CUR is also used as a photosensitizer (PS) in antimicrobial Photodynamic Therapy (aPDT). However, CUR is hydrophobic, unstable in solutions, and has low bioavailability, which hinders its clinical use. To circumvent these drawbacks, drug delivery systems (DDSs) have been used. In this review, we summarize the DDSs used to carry CUR and their antimicrobial effect against viruses, bacteria, and fungi, including drug-resistant strains and emergent pathogens such as SARS-CoV-2. The reviewed DDSs include colloidal (micelles, liposomes, nanoemulsions, cyclodextrins, chitosan, and other polymeric nanoparticles), metallic, and mesoporous particles, as well as graphene, quantum dots, and hybrid nanosystems such as films and hydrogels. Free (non-encapsulated) CUR and CUR loaded in DDSs have a broad-spectrum antimicrobial action when used alone or as a PS in aPDT. They also show low cytotoxicity, in vivo biocompatibility, and improved wound healing. Although there are several in vitro and some in vivo investigations describing the nanotechnological aspects and the potential antimicrobial application of CUR-loaded DDSs, clinical trials are not reported and further studies should translate this evidence to the clinical scenarios of infections.

## 1. Introduction

The global changes arising from globalization and climate change have a profound impact on human health, including infectious diseases [1,2]. The increased mobility of people, urbanization, greenhouse-gas emissions, pollution, deforestation, global warming, loss of sea ice, sea-level rise, extreme weather events with droughts and flood, etc., have all contributed to affect the transmission, prevalence, and spread of existing infections, such as vector-borne diseases, and the emergence of new pathogens [1,2]. In some cases, these infections have resulted in epidemics such as dengue and pandemics such as COVID-19, which the world is currently facing [3].

Notwithstanding the existence of anti-infective medications, other current concerns are the drug resistance arising from the misuse of antimicrobial agents and the emergence of multidrug-resistant species [4]. These problems are a challenge for humanity, especially when considering that the development of new drugs demands time and money. Thus, the repurposing of existing medications and alternative therapies, such as natural substances, has been investigated [5,6,7].

Curcumin (CUR) is a yellow dye (diferuloylmethane—a natural polyphenol) found in turmeric (*Curcuma longa*), which is a plant native to India and Southeast Asia. Beyond its culinary use as food flavoring and coloring, CUR also has a potential application in medicine due to its therapeutic properties, which include antioxidant, anticancer, anti-inflammatory, and antimicrobial effects [8]. CUR is not toxic and, according to the Food and Drug Administration, it is “Generally Recognized as Safe” [9]. The literature shows a plethora of studies reporting the biological and pharmacological features of CUR on health. Comprehensive reviews are available on the anticancer [10], anti-inflammatory [8], and antimicrobial [11] effects of CUR.

Nonetheless, CUR is not soluble in water, unstable in solutions, and shows low bioavailability, poor absorption, and rapid elimination from the body [12]. For these reasons, organic solvents such as ethanol, methanol, acetone, and dimethyl sulfoxide (DMSO) have been used to solubilize CUR [13]. These drawbacks hinder the in vivo use of CUR as a therapeutic agent. Thus, some approaches have been used to overcome the problems of CUR, such as the use of adjuvants and drug delivery systems. Piperine, a substance derived from black pepper, and lecithin, a phospholipid, have been associated with CUR to improve its bioavailability by blocking the metabolism of CUR and enhancing its gastrointestinal absorption [12]. Additionally, drug delivery systems have been used to solubilize CUR and protect it from degradation until it reaches the target tissue, where CUR is sustainably released [14].

Nanotechnology has been a promising field in medicine (nanomedicine). Nanoscale structures show intrinsic physical and chemical properties, which have been exploited as diagnostic and therapeutic tools [14,15]. The present study reviews the drug delivery systems (DDS) used for CUR, aiming at its antimicrobial effect. Although comprehensive reviews about the antimicrobial effect of CUR (encapsulated or not) are found elsewhere [11,16,17,18], they describe only the antibacterial and antifungal activities of CUR in DDSs [17,18]. Our review summarizes the DDSs used for CUR as an antiviral, antibacterial, and antifungal agent, encompassing different nanosystems (colloids and metals) and the relevant issues of antimicrobial resistance and the emergence of new pathogens.

## 2. Free CUR

The broad-spectrum activity of CUR as an antibacterial, antifungal, and antivirus agent was reviewed previously [11,16]. Thus, this section reviews recent studies not covered by these reviews about the antimicrobial activity of free (non-encapsulated) CUR (Table 1) before reporting the DDS used for CUR.

### 2.1. Antiviral Activity

The antiviral activity of CUR has been described against enveloped and non-enveloped DNA and RNA viruses, such as HIV, Zika, chikungunya, dengue, influenza, hepatitis, respiratory syncytial viruses, herpesviruses, papillomavirus, arboviruses, and noroviruses [11,73,74]. The action mechanism of CUR involves the inhibition of viral attachment and penetration into the host cell and interference with viral replication machinery and the host cell signaling pathways used for viral replication. Moreover, CUR works as a virucidal substance, acting on the viral envelope or proteins [11,73,74]. CUR in 0.4% vol/vol DMSO was able to inhibit several strains of the Zika virus, including those causing human epidemics, inhibiting the viral attachment to the host cell [19]. The inhibitory effect was potentiated when CUR was combined with gossypol, which is another natural product. CUR also inhibited human strains of the dengue virus [19]. The combination of CUR with heat treatment reduced the time and temperature needed for inactivating the foodborne enteric virus (hepatitis A virus and Tulane virus—a cultivable surrogate of the human norovirus) [20]. CUR was able to inhibit the lytic replication of Kaposi’s sarcoma-associated herpesvirus (KSHV) as well as reduce its pathogenesis (neoangiogenesis and cell invasion of KSHV-infected mesenchymal stem cell from the periodontal ligament) [21].

While the antiviral effect of CUR has been experimentally demonstrated, the effect of CUR against the new severe acute respiratory syndrome coronavirus 2 (SARS-CoV-2), the etiologic agent of COVID-19, has been predicted by in silico studies using computational techniques, such as molecular docking [75,76,77,78,79,80,81,82]. These in silico studies showed the binding affinity of CUR to the spike protein of SARS-CoV-2 and the human receptors of the host cell, which could inhibit the viral infection into the human cells. The targets to which CUR may bind are the viral non-structural protein 9 (Nsp9) [77] and 15 (Nsp15) [81], main proteases of SARS-CoV-2 (important for viral replication) [75,80], receptor-binding domains (RBD) of the viral spike protein [76,78,82], human cell receptors angiotensin-converting enzyme 2 (ACE2) [76,82], and glucose-regulating protein 78 (GRP78) [79], as well as the RBD/ACE complex [76]. Nevertheless, a virtual screening evaluated the interaction between potential functional foods and the main protease of SARS-CoV-2 and found that CUR showed lower docking affinity than flavonoids, vitamin, and β-sitosterol [83]. Omics approaches have been studied to identify infection pathways and propose drugs that could target these pathways. Thus, an integrative multiomics (interactome, proteome, transcriptome, and bibliome) analysis identified biological processes and SARS-CoV-2 infection pathways and proposed CUR as a potential prophylactic agent for blocking the SARS-CoV-2 infection [84].

Although most investigations have evaluated the potential of CUR against SARS-CoV-2 by computational simulations, an in vitro study showed that an immunomodulatory herbal extract composed of CUR and piperine presented a virucidal effect (viral inhibition of up to 92%) on SARS-CoV-2 [22]. Other in vitro studies showed the ability of CUR to inhibit its viral predecessor—the SARS-CoV-1 [23,24]. CUR in DMSO (<0.4%) inhibited by 25–50% the cytopathogenic effect of SARS-CoV-1 on Vero E6 cells and by 50% the viral replication and 3CL protease (main protease) [23]. Another study used CUR as a positive control for 3CL protease inhibition [24]. CUR also inhibited the papain-like protease, which is another protease used for SARS-CoV replication [25].

### 2.2. Antibacterial Activity

The antibacterial effect of CUR has been demonstrated against Gram-positive and Gram-negative species, including strains responsible for human infections and showing antibiotic resistance [11,16,85,86]. CUR also inhibits bacterial biofilms, which are communities of cells embedded in a self-produced polymeric matrix tolerant to antimicrobial treatments [11,16,85,86]. The antibacterial mechanism of action of CUR involves damage to the cell wall or cell membrane, interference on cellular processes by targeting DNA and proteins, and inhibition of bacterial quorum sensing (communication process mediated by biochemical signals that regulate cell density and microbial behavior) [85]. Moreover, CUR affected the L-tryptophan metabolism in *Staphylococcus aureus* (Gram-positive) but not in *Escherichia coli* (Gram-negative), produced lipid peroxidation, and increased DNA fragmentation in both bacteria [26]. These results, along with the increased levels of total thiol and antioxidant capacity observed after bacterial cells were treated with CUR, suggested that oxidative stress may be the mechanism of antibacterial action of CUR [26]. Therefore, these multiple targets make CUR an interesting option for antibiotic-resistant strains. CUR is effective in killing methicillin-resistant *S. aureus* (MRSA), which is a concerning pathogen responsible for nosocomial and community-associated infections [86]. CUR and another polyphenol, quercetin, inhibited the growth of MRSA and their combination was synergistic [27]. Moreover, CUR absorbs blue light (400–500 nm) and is used as a natural photosensitizer (PS) in antimicrobial Photodynamic Therapy (aPDT) [87]. CUR-mediated aPDT reduced the viability of reference strain of *S. aureus* and clinical isolates of methicillin-sensitive *S. aureus* (MSSA) and MRSA by 4 log_10_, while CUR alone reduced their survival by 2 log_10_ [28]. The aPDT mediated by CUR in 10% DMSO reduced the biofilm viability of *S. aureus* and MRSA by 3 and 2 log_10_, respectively, and their metabolic activity by 94% and 89%, respectively [29]. The antibiofilm activity of CUR-mediated aPDT was also observed against clinical isolates of vancomycin-resistant *S. aureus* (VRSA), with reductions of 3.05 log_10_ in biofilm viability, 67.73% in biofilm biomass, and 47.94% in biofilm matrix [30]. Additionally, aPDT resulted in the eradication of VRSA in a rat model of skin infection [30]. The association of CUR-mediated aPDT with artificial skin resulted in a 4.14 log_10_ reduction in *S. aureus* from infected wounds in rats [31].

The combination of CUR and another natural PS, hypocrellin B, increased the photoinactivation of *S. aureus* compared with the photodynamic effect of each PS alone [32]. Bacterial cells showed alteration in their membrane integrity and the dual-PS-mediated aPDT also decontaminated apples with *S. aureus* [32]. The CUR-mediated aPDT was also effective in decontaminating food, reducing the number of *S. aureus* recovered from meat and fruit [33]. Compared to another natural PS, aloe-emodin, CUR was less effective in photokilling *S. aureus* and *E. coli* [34]. Three-dimensional μcages fabricated with CUR and resin monomer (pentaerythritol triacrylate) polymerized by infrared light were used to entrap and kill *S. aureus*. Irradiation of μcages for 10 min with visible light resulted in a bacteria mortality rate of 95% [35]. Following the principles of aPDT, Sonodynamic Therapy (SDT) associates a PS (also called sonosensitizer) with ultrasound (US) instead of light for the treatment of deeper lesions and infections, where light cannot reach [88]. Both aPDT and SDT mediated by CUR, as well as the combination of both (SPDT, when the PS is activated by light and US simultaneously), reduced the viability of *S. aureus* biofilms. SPDT promoted the highest reduction (3.48 log_10_), which was potentiated when CUR was combined with sodium dodecyl sulfate (7.43 log_10_) [36].

Regarding Gram-negative species, CUR alone was not able to inhibit the growth of an Enterotoxigenic *E. coli*, which is a strain that causes severe diarrhea and is resistant to antibiotics [37]. However, synergism was observed between CUR at 330 μg/mL and antibiotics (Ceftazidime, Amoxicillin/Clavulanic acid, Cefotaxime, and Ampicillin) [37]. CUR did not affect the growth of enteroaggregative (EAEC) and enteropathogenic (EPEC) diarrheagenic *E. coli* but inhibited the secretion and release of their virulence factors, Pet, and EspC, which are toxins produced by these strains [38]. Conversely, CUR alone and with ampicillin inhibited the growth of other species that caused diarrhea—*Shigella dysenteriae* and *Campylobacter jejuni*, including multidrug-resistant strains [39]. The aPDT mediated by CUR and light reduced the viability of *E. coli* by 3.5 log, increased membrane permeability of bacteria, and decontaminated oysters [40]. CUR-mediated aPDT reduced the viability of *Helicobacter pylori* and its virulence factors (motility, urease production, adhesion to erythrocytes, and biofilm formation) [41]. On *Pseudomonas aeruginosa*, aPDT potentiated the inhibitory effect of CUR, inhibited biofilm formation and matrix production, reduced biofilm thickness, and downregulated quorum sensing genes [42]. The photoinactivation of imipenem-resistant *Acinetobacter baumannii* reduced bacterial viability by 97.5% and shotgun proteomics analysis identified 70 carbonylated proteins modified after CUR-mediated aPDT related to the membrane, translation, and response to oxidative stress [43].

CUR inhibited the growth of antibiotic-resistant *P. aeruginosa*, *A. baumannii*, and *Klebsiella pneumoniae* isolated from burn wound infections and showed synergism with meropenem [44]. On gastrointestinal bacteria of human and bovine origin, CUR inhibited Firmicutes (*Clostridioides difficile* and *Acetoanaerobium* (*Clostridium*) *sticklandii*) but did not affect Bacteroidetes (*Bacteroides fragilis* and *Prevotella bryantii*) [45]. CUR was conjugated to triphenyl phosphonium resulting in a compound named Mitocurcumin, which inhibited the growth of *Bacillus subtilis*, *E. coli*, *Staphylococcus carnosus*, and *Mycobacterium smegmatis*, and induced morphological changes in *B. subtilis* [46]. Seventeen synthesized monocarbonyl curcuminoids showed high antibacterial activity against MSSA and MRSA and moderate activity against *E. coli* [47]. The four most effective curcuminoids were bacteriostatic at low concentrations and bactericidal at high concentrations against MRSA, which showed membrane damage. In an ex vivo mammalian co-culture infection model, two curcuminoids decreased the viability of MSSA internalized in the fibroblasts [47]. One of thirteen synthesized curcuminoids, 3,3′-dihydroxycurcumin, showed antibacterial activity against *S. aureus*, *B. subtilis*, *Enterococcus faecalis*, and *Mycobacterium tuberculosis*, and produced membrane damage on *B. subtilis* [48]. Nonetheless, all the synthesized curcuminoids were not effective against Gram-negative species (*P. aeruginosa* and *E. coli*) [48]. CUR analogs (monocurcuminoids, MC) were synthesized and showed higher, lower, or similar antimicrobial activity than CUR against *Aeromonas hydrophila*, *E. coli*, *E. faecalis*, *K. pneumoniae*, *P. aeruginosa*, *S. aureus*, and the yeast *Candida albicans* [49]. Two MC and turmeric powder presented synergism against *A. hydrophila*, *P. aeruginosa*, and *C. albicans*. When aPDT was performed with UV light, two MC-mediated aPDT decreased the growth of *E. faecalis*, *E. coli*, and *S. aureus*, while aPDT with another MC and CUR increased the growth of *A. hydrophila*, *E. faecalis*, *S. aureus*, *C. albicans*, and *P. aeruginosa* [49]. CUR was more effective than other natural biomolecules (quercetin and resveratrol) in inhibiting the growth of *E. faecalis* in spermatozoa from rabbits, but less effective than antibiotics [50]. CUR-mediated aPDT also reduced the viability of *E. faecalis* biofilms grown in bovine bone cavities for 14 days by 1.92 log_10_ [51]. The aPDT and the combination of a nanobubble solution and the US reduced the viability of the aquatic pathogens *Aeromonas hydrophila* and *Vibrio parahaemolyticus* [52].

CUR and aPDT have been used for dental infections and oral diseases. The *Curcuma longa* extract decreased the viability of 3-week-old *E. faecalis* biofilms formed on the root canal surface of human teeth [53]. The aPDT mediated by CUR and continuous laser irradiation eradicated planktonic cultures of *Streptococcus mutans*, which is the main etiologic factor of dental caries [54]. A formulation of syrup with curcuminoids and 30% sucrose was used as a PS in aPDT, which reduced the viability of *S. mutans*, *Streptococcus pyogenes*, and a clinical isolate from a patient with pharyngotonsillitis [55]. Microbial samples from carious dentin were grown as microcosm biofilm and submitted to CUR-mediated aPDT, which reduced the vitality of 3- and 5-day-old biofilms [56]. CUR alone decreased the biomass and the viability of mono- and dual-species biofilms of *S. mutans* and *C. albicans*, as well as the production of biofilm matrix and the expression of genes related to glucosyltransferase and quorum sensing of *S. mutans*, and the adherence of *C. albicans* [57].

The therapeutic effect of CUR on periodontal diseases was extensively investigated in animal models and clinical trials, which were reviewed [89]. Beyond its antibacterial activity, CUR-mediated aPDT also produced a bystander effect (behavior change of cells exposed to treated target cells) on the periodontal pathogen *Aggregatibacter actinomycetemcomitans*, reducing its survival, metabolic activity, and the production of quorum sensing molecule [58]. The aPDT with CUR decreased the growth of both *A. actinomycetemcomitans* and *Porphyromonas gingivalis*, which is another pathogenic periodontal bacterium [59]. Antimicrobial photothermal treatment promoted higher reduction than CUR-mediated aPDT in the viability of mixed biofilms of periodontal pathogens (*P. gingivalis*, *A. actinomycetemcomitans*, *Campylobacter rectus*, *Eikenella corrodens*, *Fusobacterium nucleatum*, *Prevotella intermedia*, *Parvimonas micra*, *Treponema denticola*, and *Tannerella*
*forsythia*) grown on a titanium surface inside artificial periodontal pockets [60]. The aPDT mediated by different PS (methylene blue, CUR, and chlorin e6) eradicated the planktonic growth and reduced the biofilm viability of metronidazole-resistant bacteria from the subgingival plaque [61]. CUR alone inhibited the growth of *P. gingivalis* and CUR in gel was biocompatible when evaluated subcutaneously in rats [62].

A randomized clinical trial showed that CUR-mediated aPDT associated with scaling and root planing improved the clinical attachment level gain of periodontal pockets in type-2 diabetic patients after three and six months [63]. The aPDT with CUR and LED applied in the mouth of 30 patients with acquired immune deficiency syndrome (AIDS) reduced the counts of *Streptococcus* spp., *Staphylococcus* spp., and total microorganisms from saliva, but not the number of Enterobacteriaceae and *Candida* spp. [64]. Additionally, there was no reduction in patients with CD8 lymphocytes lower than 25% [64].

### 2.3. Antifungal Activity

The antifungal activity of CUR has been demonstrated mostly against *Candida* spp. by many in vitro and few in vivo studies [90]. CUR inhibited the growth of a reference strain and a clinical isolate of *C. albicans*, as well as reference strains of *Candida parapsilosis*, *Candida glabrata*, and *Candida dubliniensis* [65]. When biofilms of both *C. albicans* strains were evaluated, CUR reduced only the viability of the standard strain in a concentration-dependent effect, while the antifungal fluconazole did not inhibit the viability of either strain [65]. CUR and 2-aminobenzimidazole (2-ABI) inhibited the growth and adhesion of *C. albicans* and *S. aureus* to medical-grade silicone [66]. The combination of CUR and 2-ABI enhanced the inhibition of biofilm formation and reduced the viability of 48 h-old single and dual-species biofilms [66]. The aPDT mediated by CUR reduced the survival of 14-day-old biofilm of *C. albicans* in bone cavities, confirmed by fluorescence spectroscopy [67]. CUR-mediated aPDT reduced the metabolic activity of biofilms of *C. albicans* reference strain and clinical isolates from the oral cavity of patients with HIV and lichen planus [68]. Moreover, genes related to hyphae and biofilm formation were downregulated [68]. The aPDT mediated by CUR and another PS, Photodithazine^®^, also resulted in the downregulation of genes involved in adhesion and oxidative stress response in *C. albicans* biofilms [69]. CUR alone and CUR-mediated aPDT, combined or not with an antibody-derived killer decapeptide, reduced the metabolic activity of an 18 h biofilm of *C. albicans* [70]. CUR showed synergism with fluconazole and CUR-mediated aPDT inhibited the planktonic growth and reduced the biofilm viability of fluconazole-resistant *C. albicans* [71]. CUR-mediated aPDT also increased the survival of *Galleria mellonella* infected with fluconazole-susceptible *C. albicans*, but did not affect the survival of larvae infected with fluconazole-resistant strain [71]. A library of 2-chloroquinoline incorporated monocarbonyl curcuminoids (MACs) was synthetized and most of the MACs exhibited strong or moderate antifungal activity compared with miconazole against *C. albicans*, *Fusarium oxysporum*, *Aspergillus flavus*, *Aspergillus niger*, and *Cryptococcus neoformans* [72]. To suggest a possible antifungal mechanism, a molecular docking analysis showed that MACs had binding affinity to sterol 14α-demethylase (CYP51), leading to impaired fungal growth [72].

## 3. Curcumin in DDSs (Colloidal, Metal, and Hybrid Nanosystems)

### 3.1. CUR in Micelles

Micelles are aggregates of surfactants or block polymers self-assembled in water solution. They are used as DDSs and formed by a hydrophilic domain named corona and a hydrophobic domain called core (Figure 1) [91], which stays in contact with hydrophobic drugs such as CUR [91]. Micelles have low toxicity, biocompatibility, and sustained release, which makes them an attractive DDS to carry CUR and to be used in medical applications [91]. Antimicrobial studies with CUR-loaded micelles are summarized in Table 2.

CUR loaded in micelles was used against Gram-positive and Gram-negative bacteria and fungi [92]. A mixed polymer micelle was synthesized by the thin-film hydration method from the mixture of commercial Pluronic^®^ PF127 associated with P123, chitosan, sodium alginate, maltodextrin, and Tween-80. Eighteen formulations were synthesized and six showed the highest loading and entrapment efficiency of CUR (from 47.84 to 50 ppm), which were selected for the antimicrobial assays. The MIC values of CUR against *E. coli* and *S. aureus* were 350 and 275 µg/mL, respectively, although the methodology described the MIC assay of CUR loaded in micelles only for *E. coli* [92]. The antimicrobial activity of the six selected formulations was tested against *E. coli*, *S. aureus*, and *A. niger* by the agar diffusion test. Four formulations showed significant inhibition of *S. aureus* and *A. niger* compared with pure CUR, while three formulations were more effective in inhibiting *E. coli* than pure CUR. The findings reported in this study [92] suggest that micelle composition and the amount of each polymer affected the concentration of CUR in the formulation and antimicrobial activity.

Another type of micelles was synthesized using poly(ε-caprolactone)-block-poly (aspartic acid) (PCL-b-PAsp) decorated with silver (Ag) on the micelle’s corona to carry CUR [93]. Micelles were incubated with suspensions of *P. aeruginosa* and *S. aureus* for 12 h and bacterial survival was monitored by changes in optical density at 600 nm (OD600). Low OD600 values were observed for both species treated with Ag-decorated micelles with and without CUR in a concentration-dependent manner. CUR improved the antibacterial effect of Ag-decorated micelles. Bacteria stained by propidium iodide and incubated with CUR-loaded micelles showed less fluorescence, while strong red fluorescence was observed with Ag and Ag-CUR micelle, suggesting that Ag nanoparticles on micelles were responsible for disrupting the bacterial membrane. Another interesting finding was the fast CUR release from micelles incubated with *Pseudomonas* lipase (a virulence factor in *P. aeruginosa* infections), which promoted the degradation of PCL on micelles, enabling CUR release [93].

An important mechanism of antimicrobial resistance is the overexpression of efflux pumps, which are membrane transporters of the microbial cell responsible for expelling out toxic substances. A polymer formed by monomethoxy poly (ethylene glycol) linked to oleate (mPEG-OA) was used to encapsulate CUR extracted from the rhizome of *Curcuma longa*, in micelles/polymersome nanoparticles (CMN) [94]. Loaded micelles were used against four ciprofloxacin-resistant clinical isolates of *P. aeruginosa* to evaluate the efflux pump inhibition of CUR. Initially, the MIC of ciprofloxacin against each clinical isolate was determined (range from 8 to 1024 µg/mL), then the antimicrobial activity of ciprofloxacin at sub-MIC associated to CMN at 0, 10, 25, 75, 100, 250, 400, 500, 750, and 1000 µg/mL was evaluated by broth microdilution. CMN at 400 µg/mL associated with ½ MIC ciprofloxacin for 24 h promoted 50% of bacterial death. Afterward, CMN at 400 µg/mL associated with sub-MIC ciprofloxacin showed significant downregulation of the efflux-pump genes mexX and oprM compared with ciprofloxacin alone. CMN worked as an efflux pump inhibitor, then, ciprofloxacin could enter inside the bacteria. To confirm the uptake of ciprofloxacin in bacterial cells, the association of CMN at 400, 800, or 1200 µg/mL with ¼ or ½ MIC of ciprofloxacin for 24 h resulted in increased absorption values of ciprofloxacin in cell lysate with higher concentrations of CMN [94].

The association of CUR in micelles with a commercial antifungal agent (ketoconazole, KCZ) was reported against *C. albicans* [95]. CUR in PEG-PCL micelles (CUR-M) at 256 µg/mL was required to inhibit fungal growth by eighty percent (MIC80). Additional micelles were synthesized to carry KCZ (KCZ-M), which showed an MIC80 of 8 µg/mL. Moreover, the combination of both drugs promoted a synergistic effect (2.8 and 0.5 µg/mL of CUR and KCZ, respectively). Higher inhibition zones were observed for samples incubated with micelles containing both drugs (KCZ-CUR-M), but the method used to encapsulate both drugs on micelles was not reported. The effect of micelles on 48 h biofilms showed reductions in fungal viability of 2.66% and 17.7% after incubation for 48 h with KCZ-M and CKZ-CUR-M. Confocal microscopic images showed the strongest disruption in biofilms treated with the combination of CUR-M and KCZ-M [95].

The antimicrobial effect of SDT using CUR in micelles associated with the US at 1.56 W/cm^2^ for 1 min was evaluated against the cariogenic bacteria *S. mutans* [96]. CUR-loaded micelles were synthesized using PEG-α phosphatidylethanolamine (PEG-PE) with a final concentration of CUR at 50 mM and showed a spherical shape, an average diameter of 22 nm, and an anionic charge (−17.3 mV). The uptake of free CUR and CUR-loaded micelles by *S. mutans* cells increased from 5 min to 1 h. The CUR-loaded micelles associated with the US promoted higher bacterial sonoinactivation (99.9%) compared to SDT mediated by free CUR (90.8%). Moreover, free CUR and CUR-loaded micelles alone reduced the bacterial CFU/mL by 23.4% and 47.2%, while the US alone reduced the bacterial viability by up to 11.2%. These results were confirmed by fluorescence microscopy. Additionally, the production of Reactive Oxygen Species (ROS) was increased by 7.3- and 10.8-fold in *S. mutans* suspensions subjected to SDT mediated by free CUR and CUR-loaded micelles, respectively [96].

Three different CUR-loaded micelles were synthesized with a common C16-long hydrocarbon chain, as the hydrophobic length that was linked to three different cationic amines (hydrophilic portion): N,N-di-(3-aminopropyl)-N-methylamine (DAPMA), spermidine (SPD), and spermine (SPM) [97]. The antimicrobial effect of micelles was compared with free CUR against planktonic cultures of *P. aeruginosa*. Free CUR at 50 µM reduced bacterial growth by nearly 50%, while CUR in micelles of SPD at 1 µM, DAPMA at 500 nM, and SPM at 250 nM eradicated bacterial growth [97]. Because CUR in SPM was the most effective nanomicelle, it was selected to evaluate its antibacterial photodynamic effect at sub-inhibitory concentrations (50 and 100 nM) associated with a blue laser light (445 nm) at fluences of 6, 18, and 30 J/cm^2^. Although free CUR at 5–200 µM with light did not show an antibacterial photodynamic effect, both concentrations of CUR in SPM micelles associated with laser at 18 and 30 J/cm^2^ eradicated bacterial growth [97].

The aPDT mediated by CUR in commercial Pluronic^®^ P123 at different pHs (5, 7.2, and 9) was evaluated against planktonic cultures of *S. aureus*. Bacterial suspensions were incubated with CUR/P123 in the dark for up to 30 min and illuminated for up to 30 min. Complete bacterial eradication was observed with 7.80 μmol/L CUR in P123 at a pH of 5.0 and 6.5 J/cm^2^ light, while pHs of 7.2 and 9.0 showed reductions in bacterial viability of 1.5–2 log and 1–1.5 log, respectively. A mathematical model showed that the incubation time with CUR/P123 in the dark before irradiation did not affect bacterial viability at a pH of 7.2, while at pHs of 5.0 and 9.0, the incubation in the dark affected bacterial photoinactivation but without a significant reduction [98].

A cationic micelle was synthesized with methoxyl poly(ethylene glycol)-*block*-poly(ε-caprolactone) (PEG- b-PCL) and poly(ε-caprolactone)-*block*-poly(hexamethylene guanidine) hydrochloride-*block*-poly(ε-caprolactone) (PCL-b- PHMG-b-PCL), which was mixed with the anionic sodium carboxymethyl cellulose to produce a shear-thinning electrostatic (STES) hydrogel capable of being injected through a syringe. The antibacterial effect of STES hydrogel was demonstrated by the inhibition zone test against *S. aureus* and *E. coli* and by minimum inhibitory concentration (MIC), which resulted in values of 16 and 32 μg/mL, respectively. CUR was encapsulated in the core of micelles and the STES hydrogel was then evaluated as an antibacterial wound healing in rats. After 14 days, rats treated with CUR in STES hydrogel showed a higher reduction in the wound area, the recovery of *S. aureus* from wounds, and better histological features of healing than animals treated with STES hydrogel without CUR and untreated control rats [99].

Micelles of poly-(lactic-co-glycolic acid) (PLGA) and dextran (DEX) inhibited the growth of *Pseudomonas fluorescens* [100]. The functionalization of the dextran shell with CUR resulted in cationic micelles, which increased the antibacterial effect against *P. fluorescens* and inhibited *Pseudomonas putida* in a concentration-dependent manner, whereas free CUR showed no antibacterial activity. CUR-PLGA-DEX micelles also presented antibiofilm activity against both bacterial species, inhibiting biofilm formation and disrupting preformed biofilms. While CUR-PLGA-DEX micelles did not decrease the biofilm viability of *P. fluorescens*, the viability of *P. putida* biofilms was reduced to up to 25%. Moreover, confocal microscopy showed that the micelles were able to penetrate in the *P. putida* biofilm but not in the *P. fluorescens* biofilm [100].

CUR in micelles (nanoCUR) was also evaluated on the modulation of inflammatory cytokines in COVID-19 patients [101]. Forty patients diagnosed with COVID-19 and 40 healthy controls were selected. The COVID-19 patients received 160 mg of nanoCUR or placebo (*n* = 20 each group) in 40 mg capsules, four times per day, for 14 days. Blood samples were collected from each patient before and after the interventions and the mRNA expression and the secretory level of inflammatory cytokines were evaluated by real-time PCR and ELISA, respectively. All COVID-19 patients showed higher cytokine expression and secretion levels compared with the healthy subjects. The patients treated with nanoCUR showed a decrease in the mRNA expression of interleukin-1β (IL-1β) and interleukin-6 (IL-6) compared with their baseline values and the placebo group, but there was no reduction in interleukin-18 (IL-18) and Tumor Necrosis Factor-α (TNF-α) compared with the placebo group. Compared with the baseline values, nanoCUR reduced the secretory level of all cytokines from the supernatant of cultured cells and IL-1β, IL-6, and TNF-α from serum. Compared with the placebo group, reductions in IL-6 from serum and IL-1β and IL-6 from the supernatant were observed in the nanoCUR group. Moreover, in the nanoCUR groups, patients showed improvement of fever, cough, and dyspnea [101]. Nonetheless, the viral load in COVID-19 patients was not measured, thus the antiviral effect of nanoCUR on SARS-CoV-2 was not evaluated.

### 3.2. CUR in Liposomes

Liposomes are biodegradable and biocompatible systems, which consist of hydrophobic and hydrophilic groups (Figure 2) [102]. The hydrophobic layer is mainly composed of phospholipids and cholesterol molecules. This lipid-based carrier is suitable for administering water-insoluble drugs, such as CUR [103]. Liposomes are classified into three groups: single unilamellar vesicles, large unilamellar vesicles, and multilamellar vesicles [104]. Drugs encapsulated in liposomes are protected from chemical degradation and show increased drug solubility [102]. Additionally, liposomes have advantageous properties such as better penetration into the skin, deposition, anti-melanoma, and antimicrobial activity [103]. Antimicrobial studies with CUR-loaded liposomes are summarized in Table 3.

CUR in liposome was prepared with lecithin and cholesterol and its inhibitory effect on the quorum sensing (QS) of *Aeromonas sobria* was determined [105]. The MICs of CUR in liposomes were 420, 400, and 460 μg/mL against *A. sobria*, *Chromobacterium violaceum*, and *Agrobacterium tumefaciens*, respectively, while the MICs of free CUR were 280, 200, and 370 μg/mL, respectively. *C. violaceum* and *A. tumefaciens* detected the QS molecules N-acylhomoserine lactones (AHLs) produced by *A. sobria*. At sub-MIC, CUR in liposomes inhibited violacein production and the β-galactosidase from *C. violaceum* and *A. tumefaciens*, respectively. CUR in liposomes at sub-MIC also inhibited the QS-controlled swimming and swarming motility, siderophore production, and extracellular proteases of *A. sobria*. CUR in liposome inhibited the biofilm formation of *A. sobria* by up to 93.35%, which was confirmed by scanning electron microscopy (SEM) images. Confocal microscopy images showed reduced thickness of *A. sobria* biofilms treated with CUR in liposomes. CUR in liposomes also reduced the AHL production of *A. sobria* by up to 93.55%. The in silico analysis suggested that CUR from liposomes interacted with Luxl protein of *A. sobria*, which inhibited bacterial QS [105].

PEGylated encapsulated CUR nanoliposomes (PCNL) were synthesized, characterized, and evaluated against bacteria (*B. subtilis*, *K. pneumoniae*, *C. violaceum*, *E. coli*, and acid-resistant bacteria *M. smegmatis*) and fungi (*A. niger*, *C. albicans*, and *Fusarium oxysporum*) using a disc diffusion assay [106]. PCNL produced higher or similar inhibition zones than native CUR against all species. Higher inhibition zones were found against *M. smegmatis*, *C. violaceum*, and *A. niger*. Moreover, the reduced production of violacein suggested that PCNL inhibited the QS activity of *C. violaceum* [106].

Liposome formulations containing CUR were synthesized with phospholipids of different lengths of acyl chains, superficial charge (anionic and cationic), and presence or absence of cholesterol [107]. The lipid composition and liposomes charge affected the physicochemical characterization of liposomes, such as entrapment efficiency, fluorescence quenching, and antioxidant activity. Only cationic liposomes with CUR inhibited the growth of *S. aureus* with a MIC value of 7 μg/mL in the range 3.5−14 μg/mL. The presence or absence of cholesterol did not change the antibacterial effect [107].

Nanocapsules of poly-(lactic-co-glycolic acid) (PLGA) and Pluronic F68 (poloxamer 188) containing CUR in their oily (medium-chain triglycerides) core were evaluated against Gram-negative (*E. coli*, *Salmonella typhimurium*, and *P. aeruginosa*) and Gram-positive (*S. aureus*, *Bacillus sonorensis*, and *Bacillus licheniformis*) bacteria [108]. CUR in nanocapsules was more effective than free CUR (CUR in DMSO and distilled water) against all species evaluated, considering that lower MIC values were observed for CUR in nanocapsules (75 and 100 μg/mL) than for free CUR (100 to 300 μg/mL). Gram-positive species were more susceptible to CUR in nanocapsules than Gram-negative species, with MIC values of 75 and 100 μg/mL, respectively [108].

The encapsulation of CUR and berberine (another phytochemical) in liposomes reduced their MIC against MRSA compared with their free forms [109]. The co-encapsulation of both drugs in liposomes (BCL) showed synergism and increased the bacteriostatic effect compared with clindamycin. The cytotoxicity assay showed that the inhibitory concentration of BCL against MRSA was 5-fold lower than that observed for mammalian cells (fibroblasts). When MRSA biofilms were evaluated, co-encapsulated drugs inhibited biofilm formation, reducing their biomass and viability. When fibroblasts were infected with MRSA, treatment with BCL was superior to clindamycin in inhibiting the intracellular infection and reducing the viability of bacteria recovered from mammalian cells. BCL also reduced the levels of inflammatory cytokines (IL-6 and TNF-α) produced by differentiated macrophages infected with MRSA [109].

In addition to its antimicrobial effect, the in vivo antimalarial effect of CUR in liposomes has also been demonstrated. CUR in liposomes combined or not with antimalarial drugs decreased the parasitemia levels and increased the survival of mice infected with *Plasmodium berghei* [102,111] and *Plasmodium yoelii* [112].

### 3.3. CUR in Solid Lipid Nanoparticles

Solid lipid nanoparticles (SLN, Figure 3, Table 3) are a modern type of lipid-based carrier composed by solid biodegradable lipids and spherical solid lipid particles. SLNs are water colloidal or aqueous surfactant solution systems [103]. SLNs have advantages such as biocompatibility, biodegradability, greater drug absorption, and drug retention [18,103], thus they are an interesting system to carry CUR [14]. Currently, SLNs have become popular because they are used as carriers for COVID-19 vaccines based on RNA vaccine technology (Moderna and Pfizer–BioNTech).

The antimicrobial activity of CUR in solid lipid nanoparticles (CurSLN) was evaluated, aiming at the treatment of oral mucosal infection [110]. Compared with other lipid-based colloidal nanocarriers (liposomes and nanoemulsions), solid lipid nanoparticles have a rigid morphology, which can be an advantage for topical treatments. The mucoadhesive properties of two CurSLN formulations composed of lipid stabilizer (Gelucire 50/13-Gelucire 50/13 and Gelucire 50/13-poloxamer 407) and CUR suspension were evaluated by CurSLN adhesion to mucin and ex vivo mucoadhesion, permeation, and retention tests using chicken buccal mucosa. The CurSLN composed by Gelucire 50/13-poloxamer 407 showed the highest mucoadhesive properties and was therefore selected for the antimicrobial assays against Gram-positive species (*S. aureus*, *S. mutans*, *Viridans Streptococci*, *Lactobacillus acidophilus*), Gram-negative species (*E. coli*), and fungus (*C. albicans*). The CurSLN showed the MICs values ranging from 0.09375 to 3 mg/mL, which were increased from twofold to eightfold compared with CUR powder and CUR stabilized in poloxamer 407. Additionally, only CurSLN showed either a microbicidal effect (no colony growth) or an 80% reduction in colony growth for Gram-positive species and fungus. Conversely, there was no reduction for *E. coli*, indicating a lower efficacy of CUR against Gram-negative species. The CurSLN showed mucoadhesive features and antimicrobial activity [110].

### 3.4. CUR in Nanoemulsions

Nanoemulsions (NE) are thermodynamically stable dispersions of oil and water (Figure 4) [113]. They are formed by a phospholipid monolayer composed of a surfactant and co-surfactant, which are important for nanoemulsion stabilization [113,114]. This system has thermodynamic stability and high solubilization characteristics, with improved drug release kinetics [115]. NE systems can be manufactured through emulsification, which can control the size of the drops and increase the drug solubility and efficacy. Moreover, the main disadvantage of NE is the high amount of surfactants in the formulation, which can lead to a potential toxic effect [114]. Antimicrobials studies with CUR-loaded NE are summarized in Table 4.

Tetrahydrocurcumin (THC), which is a colorless CUR derivative, was evaluated as an anti-HIV agent for vaginal application [116]. An in silico analysis showed that CUR and THC have similar gp120-CD4-binding inhibitory activity, suggesting a potential anti-HIV effect. In vitro results showed that both THC and CUR had anti-HIV activity at lower concentrations (3.639 and 4.372 μM, respectively) than their cytotoxic concentrations observed in TZM-bl cells (539 and 591 μM, respectively). Microemulsions (ME) were then synthesized after optimizing the composition of oil, surfactant/co-surfactant, and water. THC at 5% was loaded in oil-in-water ME and characterized. The cytotoxic concentration of THC-loaded ME was <564 μM against different cell lines. The anti-HIV activity of THC-loaded ME was 4-fold higher (0.9357 μM) than free THC (3.639 μM). THC-loaded ME also inhibited the antigen P24 of HIV-1 SF162 by more than 50%. At concentrations up to 1000 μM, THC-loaded ME did not inhibit the viability of *L. acidophilus* and *Lactobacillus casei*, which are species found in the vaginal microbiota. A gel formulation of THC-loaded ME was prepared and characterized and presented slow and complete permeation across 0.22 μm nylon membrane of the Franz diffusion cell, suggesting that the formulated gel could be used as a coitus-independent formulation. The formulated gel also kept the antiviral effect of THC-loaded ME [116].

The aPDT mediated by CUR in NE showed antiviral activity against variants of human papillomavirus 16 (HPV-16) [117]. HPV DNA was detected in biopsies of vulvar intraepithelial neoplasia and HPV types and variants were identified by genotyping and sequencing. The A431 cell line was transduced with the E6 gene from the HPV-16 variants E-P and E-350G. Transduced cells showed an uptake of CUR in NE, which was less toxic than free CUR. aPDT mediated by CUR in NE reduced cell viability by over 85% and increased the activity of caspases 3 and 7 in the apoptosis assay. Cells grown as organotypic cultures, resembling epithelial tissues, and submitted to CUR in NE-mediated aPDT showed fragmented morphology, suggesting tissue damage [117].

The antiviral activity of CUR-loaded NE was observed against four dengue virus serotypes (DENV-1 to -4) [118]. CUR-loaded NE showed higher toxicity against the A549 cell line than free CUR, although concentrations up to 12.5 µg/mL resulted in cell viability higher than 80%. Therefore, the antiviral experiments were performed with CUR-loaded NE and free CUR at concentrations of 1, 5, and 10 µg/mL. CUR-loaded NE and free CUR decreased the dengue virus titer. Higher reductions were observed for DENV-1 and DENV-2 than for DENV-3 and DENV-4. The viability of A549 cells infected with DENV-1, -2, and -3 was higher than 80%, while the viability of cells infected with DENV-4 was near 70% [118].

The antibacterial effect of CUR and Polyphenon 60 (P60), which is a mixture of green tea catechins, encapsulated in NE was evaluated against *E. coli* [119]. CUR and P60 showed MIC values of 0.3 and 3.3 mg/mL, respectively, and synergistic interaction. Optimized parameters were established for CUR and P60-NE synthesis using a prediction model evaluating mean droplet size, zeta potential, and polydispersity index. CUR and P60 in NE-based gel (NBS) were then prepared using chitosan and glacial acetic acid. CUR+P60 in NBS inhibited the growth curve of *E. coli* at 5 h of incubation, while the aqueous solution of CUR and P-60 inhibited bacterial growth at 15 h. The in vivo drug accumulation and biodistribution were evaluated in female rats using radiolabeled P60 with CUR in NBS. Gamma scintigraphy presented drug accumulation in the kidney and urinary bladder after 24 h of intravaginal administration of radiolabeled P60+CUR NBS, but no accumulation after oral administration. The aqueous form of radiolabeled P60+CUR was not detected after oral and intravaginal administration. Biodistribution studies confirmed the scintigram findings [119].

CUR and coumarin, which is a natural substance with pharmaceutical applications, were encapsulated in Pickering oil-in-water emulsion stabilized by aminated cellulose nanoparticles (ANC) [120]. Using solid particles instead of toxic surfactants at the oil-water interface, Pickering emulsions (PE) are considered more stable over time. NE was first synthesized changing the experimental factors (wt% of coconut oil as medium-chain triglyceride, wt% of tween 80 as a non-ionic surfactant, and temperature of oil phase). The optimal conditions were established by evaluating the average particle diameter and polydispersity index. Then, PE was synthesized by adding ANC to the organic phase. CUR or coumarin was added to the oil phase for their encapsulation. After PE characterization, its antimicrobial activity was evaluated against *S. aureus*, *S. epidermidis*, *Staphylococcus faecalis*, *E. coli*, and *C. albicans* by the agar diffusion method. CUR-loaded PE showed average inhibition zones of 9.81, 14.00, 12.80, 9.88, and 10.69 mm for the species mentioned, respectively, and the values found for coumarin-loaded PE were 10.01, 9.90, 11.84, 8.45, and 9.83 mm, respectively. Gram-positive species were more susceptible to both CUR- and coumarin-loaded PEs. Moreover, the maintenance inhibition ratio was calculated for both PE with bioactive compounds. The highest values were found against *S. epidermidis* (89% and 85% for CUR- and coumarin-loaded PEs, respectively) and the lowest values were verified against *E. coli* (62% and 59%, respectively) [120].

Seven CUR-loaded self-emulsifying drug delivery systems (CUR-SEDDS) were synthesized and evaluated against *Leishmania tropica* and bacteria isolated from cutaneous leishmaniasis [121]. The bacteria were identified by biochemical tests as *S. aureus*, *P. aeruginosa*, *E. coli*, and *K. pneumoniae*. The MIC of CUR-SEDDS (45–62 μg/mL) was lower than the MIC of free CUR (152–195 μg/mL) for all bacterial species evaluated. CUR-SEDDS showed higher antileishmanial activity than free CUR. In the cytotoxicity assay, CUR-SEDDS caused 1–2% hemolysis of the erythrocytes from human blood. In an ex vivo skin model, two CUR-SEDDS diffused on damaged skin of porcine animals [121].

CUR in NE of omega-3 rich oils (flaxseed or fish) was incorporated in polymeric microbeads of chitosan, alginate, or both polymers [122]. Their antibacterial activity was evaluated against Gram-negative (*E. coli*, *S. typhimurium*, *Yersinia enterocolitica*, and *P. aeruginosa*) and Gram-positive (*S. aureus*, *Bacillus cereus*, and *Listeria monocytogenes*) species by the agar well diffusion assay. Fish oil in alginate microbeads with and without CUR showed the highest inhibition zones (14.5–21 mm) against all bacterial species. Comparing the chitosan microbeads, flaxseed oil in chitosan (0–23 mm) had superior antibacterial activity than fish oil in chitosan (0–17.3 mm). Fish oil in chitosan and alginate microbeads showed higher inhibition of Gram-positive species (0–18.3 mm) than flaxseed oil in combined microbeads (0–13 mm). When CUR was incorporated in NE, fish oil in chitosan, alginate, and their combination showed improved antibacterial activity against Gram-negative species (0–20 mm) but reduced activity against Gram-positive species (0–18 mm). The presence of CUR in flaxseed oil incorporated in microbeads reduced the antibacterial effect against all species (0–15.5 mm) [122].

Oil-in-water high internal phase emulsion (HIPE) was prepared with sulfomethylated lignin and alkyl polyglucoside as an emulsifier under neutral conditions [123]. The sulfonation degree, the lignin concentration, and the oil/water ratio affected the size of the emulsion droplet and the stabilization of HIPE. CUR-loaded HIPE showed resistance to UV irradiation and oxidation and inhibited the growth of *S. aureus* at concentrations above 2.4 mg/mL [123].

The properties of oil-in-water NE stabilized by N-oxide surfactants (double-head—single-tail and single-head—single tail) in different oil (isooctane IO, isopropyl myristate IPM, or glyceryl monocaprylate GM) showed that IPM and GM produced NE with the smallest droplet size and polydispersity index [124]. Both surfactants resulted in stable NE, although the NE of single-headed one-tail surfactant was slightly more stable with IPM and GM. NE at concentrations up to 1.25 μL/mL was not toxic to human dermal fibroblasts. The NE of double-headed surfactant presented higher ex vivo penetration on the skin than NE of one-headed surfactant and both NE did not change the epithelial stratum corneum. NE of either surfactants or GM showed antifungal activity against *C. albicans* and the encapsulation of CUR in NE of double-headed surfactant increased its antifungal effect, although the highest reduction in fungal viability was observed for NE of single-headed surfactant with or without CUR. The antibiofilm activity of dressing CUR-loaded NE showed a reduction in *C. albicans* biofilm by 37–80%, while free CUR reduced the fungal biofilm by 30% [124].

### 3.5. CUR in Cyclodextrin

Cyclodextrins (CDs) have revolutionized the pharmaceutical industry in recent years [125]. CDs consist of three naturally occurring oligosaccharides in a cyclic structure produced from starch [126,127,128]. The natural CDs have their nomenclature system and their chemical structure based on the number of glucose residues in their structure: 6, 7, or 8 glucose units, which are denominated α-CD, β-CD, and γ-CD, respectively [129,130]. Although the entire CD molecule is soluble in water, the interior is relatively non-polar and creates a hydrophobic microenvironment. Therefore, CDs are cup-shaped, hollow structures with an outer hydrophilic layer and an internal hydrophobic cavity (Figure 5) [129]. They can sequester insoluble compounds within their hydrophobic cavity, resulting in better solubility and consequently better chemical and enzymatic stability [127]. Due to the cavity size, β-CD forms appropriate inclusion complexes with molecules with aromatic rings [131], such as CUR [132]. Antimicrobial studies with CUR in CDs are summarized in Table 5.

The main advantages of CDs include the production from a natural substance (starch), thus being a sustainable technology to the environment; positive cost–benefit ratio; low/no cytotoxicity; biodegradable and biocompatible substance [143]. CDs have been used with antifungal drugs as topical, oral, and intravenous formulations, improving the antifungal effect by synergism or mimicking the antifungal mechanism of action [126]. The antibacterial effect of aPDT mediated by CUR in methylated β-CD or polyethylene glycol-based β-CD or γ-CD was evaluated against *E. coli* and *E. faecalis* [133]. Supersaturated solutions of CUR in CD showed high photoinactivation of less susceptible Gram-negative *E. coli*. Storing CUR solutions (methylated β-CD and γ-CD polyethylene glycol) for 24 h showed a photoinactivation similar to fresh preparations. The photoinactivation of *E. faecalis* was also observed for all CUR preparations and aPDT mediated by CUR in methylated β-CD resulted in complete eradication of bacterial viability [133].

The solid dispersion of CUR was synthesized through the lyophilization of supersaturated CUR in hydroxypropyl-β-CD stabilized by hydroxypropyl-methylcellulose (HPMC) [134]. The CUR concentration of the dissolved lyophilizates decreased by 90% after storage for 168 h. Therefore, 2-day-old and fresh solutions were used for aPDT. No colony of *E. coli* was observed after aPDT using formulation with 25 μM CUR and light fluences of 14 and 28 J/cm^2^ [134]. Another formulation was prepared using CUR in methyl-β-CD, hyaluronic acid, and HPMC as precipitation inhibitors [135]. The aPDT mediated by this formulation resulted in the eradication of *E. faecalis* at CUR concentrations of 0.5 μM combined with 11 J/cm^2^ of light fluence. A complete photoinactivation of *E. coli* was observed at CUR concentrations of 5 and 10 μM combined with 32 and 16 J/cm^2^, respectively [135].

The inclusion complex of CUR in carboxymethyl-β-CD was used to develop an antimicrobial coating over the biomaterial polyethylene terephthalate (PET) [136]. The polyelectrolyte, multilayer coating, comprised positively-charged poly-L-lysine (PLL) and negatively-charged poly-L-glutamic acid (PLGuA) deposited alternately over the PET surface, which was cross-linked or left unbound (native samples). The tetralayer deposition (PLL-PLGuA-PLL-CD) was repeated 5, 10, or 20 times and CUR was loaded at 20 μM. The antimicrobial activity against *E. coli* showed that 20 multilayer coatings had the highest killing rates, which were potentiated by light irradiation (white light). While a reduction of 3.3 log (CFU/cm^2^) was observed for 20 multilayer coatings on native samples kept in the dark for 24 h, the irradiation for 8 h increased the log reduction by 5 log. Additionally, the cross-linked samples did not show significant differences from native samples, which suggested that the phototoxic effect of cross-linked coatings was limited to the outermost layers [136].

A hydrogel wound dressing based on bacterial cellulose loaded with inclusion complexes of CUR with hydroxypropyl-β-CD was synthesized and its antibacterial activity was evaluated with the disc diffusion assay [137]. The hydrogel showed inhibition zones of 11.8 ± 0.9 mm against *S. aureus*. Moreover, the hydrogel also presented hemocompatibility, cytocompatibility, and antioxidant effect [137].

Inclusion complexes of CUR-C3 (combination of 75% CUR, 16% demethoxycurcumin, and 4% bisdemethoxycurcumin) in β-CD showed MICs of 0.25 and 0.31 mg/mL against *E. coli* and *S. aureus*, respectively [138]. The growth inhibition of *E. coli* observed for native CUR-C3 was higher than those for CUR-C3 in α- and β-CD. For *S. aureus*, CUR-C3 in α-CD at a molar ratio of 1:3 and 1:5 presented higher growth inhibition than native CUR-C3, while the bacterial inhibition observed for CUR-C3 in β-CD was similar to native CUR-C3. Moreover, native CUR-C3 showed a higher antibacterial effect than native CUR against both species. Compared with native CUR, CUR in α- and β-CD at a molar ratio of 1:1 increased the inhibition of *E. coli*, while CUR alone in α-CD at a molar ratio of 1:1 was able to increase the inhibition of *S. aureus* [138].

CUR in β-CD and γ-CD loaded in chitosan (CS) presented antibacterial effects against *S. aureus* and *E. coli* [139]. For both bacteria, chloramphenicol had the highest inhibition zone, followed by free CUR, CUR in γ-CD, CUR in β-CD, CUR in γ-CD loaded in CS, and CUR in β-CD loaded in CS. The MICs of CUR in γ-CD loaded in CS and CUR in β-CD loaded in CS were 64 and 32 μg/mL, respectively, while the MIC of free CUR was 64 μg/mL for both bacteria. The formulations also showed antioxidant activity and increased CUR release at an acidic pH [139].

The antifungal effect of a formulation of γ-CD with curcuminoids (65–82% CUR, 15–25% demethoxycurcumin, and 2–7% bisdemethoxycurcumin) was evaluated against six strains of the dermatophyte *Trichophyton rubrum* isolated from active tinea lesions [140]. The MIC of terbinafine was <0.06 mg/L while the MIC of itraconazole ranged from ≤1 to 8 mg/L for all strains. The formulation showed moderate growth inhibition of five strains and strong inhibition of one strain. When the formulation was used as a PS, aPDT resulted in complete growth inhibition of all strains. No mycelium was observed in samples submitted to aPDT [140].

The inclusion of CUR and piperine in 2-hydroxypropyl-β-CD as a nutraceutical system increased the solubility of CUR and piperine by 48-fold and twofold, respectively [141]. A parallel artificial membrane permeability assay simulating in vitro the gastrointestinal wall and blood–brain barrier showed that the permeability of CUR and piperine in the nutraceutical system increased almost thirty and four times, respectively. The antioxidant properties of CUR and piperine increased in the inclusion complex. In the agar diffusion test, CUR and piperine in 2-hydroxypropyl-β-CD had increased antibacterial activity against *Clostridium butyricum*, *Streptococcus pyogenes*, *L. monocytogenes*, *B. subtilis*, *S. aureus*, and *P. aeruginosa*, but a decreased effect against *E. coli* and *E. faecalis* compared with the values found for either CUR or piperine alone. Against *Proteus mirabilis*, *Enterobacter aerogenes*, *C. difficile*, *K. pneumoniae*, *S. typhimurium*, *C. albicans*, and *Candida krusei*, the nutraceutical system showed antimicrobial activity similar to CUR or piperine alone [141].

The antibacterial activity of two CUR formulations (methyl-β-CD and polyelectrolyte-coated monolithic nanoparticles (CNP)) against *E. coli* was compared [142]. Bacterial cells incubated with CUR in methyl-β-CD showed higher uptake than those incubated with CUR in CNPs. The MIC of CUR in methyl-β-CD (500 μM) reduced bacterial viability by 2.6 log_10_ and 3 log_10_ of reduction was observed at a concentration of 625 μM. Upon irradiation, the MIC of CUR in methyl-β-CD reduced to 90 μM and resulted in a bactericidal effect. In turn, CUR in CNPs resulted in a bacteriostatic effect even when combined with light. The antibacterial mechanism of CUR in methyl-β-CD was correlated with electron transport activity and ROS production, while the mechanism of CUR in CNPs was correlated with the disruption of membrane potential and depletion of ATP content. Moreover, CUR in CNPs also resulted in the filamentation of the cells [142].

### 3.6. CUR in Chitosan

Chitin is a natural polysaccharide commonly found in the exoskeleton of marine crustaceans such as shrimps, prawns, lobsters, and crabs. Chitosan (CS) derives from the acetylation of chitin and has a linear structure of D-glucosamine (deacetylated monomer) linked to N-acetyl-D-glucosamine (acetylated monomer) through β-1,4 bonds [144]. The main advantages that make CS a promising drug carrier include biocompatibility, biodegradability, non-toxicity, controlled release system, mucoadhesive properties, and low cost [144,145]. Moreover, CS is soluble in aqueous solutions and is the only pseudo-natural polymer with a positive charge (cationic) [146], which can interact with negatively-charged DNA, membranes of microbial cells, and biofilm matrix [147]. Antimicrobial studies with CUR in CS are summarized in Table 6.

CUR in nanoparticles (NP) of CS and PEG 400 inhibited the growth of MRSA and *P. aeruginosa* after 8 h of incubation [148]. CUR in NPs reduced the colony growth of MRSA and *P. aeruginosa* by 97% and 59.2%, respectively. Transmission electron microscopy (TEM) images showed edema, distortion, and lysis of MRSA cells in contact with CUR in NPs. Infected burn wounds of mice treated topically with CUR in NPs, once a day, for seven days showed a reduction in MRSA counts compared with untreated infected controls and accelerated wound healing. Moreover, CUR in NPs did not show in vitro and in vivo toxicity against keratinocytes and embryonic zebrafish [148].

CUR-conjugated CS microspheres (CCCM) were synthesized, characterized, and evaluated against *E. coli* and *S. aureus* [149]. CCCM resulted in a higher inhibition zone than CUR alone for both *E. coli* (28 and 23 mm, respectively) and *S. aureus* (33 and 31 mm, respectively). CCCM reduced bacterial viability in a concentration-dependent manner. Additionally, CCCM presented in vitro antioxidant and anti-inflammatory effects, hemocompatibility, and no cytotoxicity against fibroblasts [149].

The MIC of CUR-loaded CS nanoparticles against the cariogenic bacterium *S. mutans* (0.114 mg/mL) was lower than the MIC obtained for CUR in nanoparticles of alginate or starch (0.204 mg/mL) [150]. All CUR-loaded nanoparticles inhibited biofilm formation by 89.48–99.38% at pHs of 5 and 7, while 59.97 and 67.38% of inhibition were observed for free CUR. CS nanoparticles showed higher CUR release than alginate and starch nanoparticles [150].

CUR was entrapped in a nanocomposite of CS, carboxymethyl starch, and montmorillonite clay (CS-CMS-MMT), which is used to strengthen the polymeric nanosystem [151]. The MIC of CUR-loaded CS-CMS-MMT against *S. mutans* was 0.101 mg/mL, while the MICs of free CUR in ethanol and water were 0.475 and 1.53 mg/mL. CUR in nanocomposite also reduced the biofilm biomass of *S. mutans* by 95.5 and 93.7% at pHs of 5 and 7, respectively, while reductions of 67 and 60%, respectively, were observed in the biofilm biomass for free CUR in ethanol. The release of CUR from the nanocomposite was higher and faster at a pH of 4.5 than 7.4 [151].

A thermosensitive hydrogel of CS and β-glycerophosphate loaded with inclusion complex of CUR and β-CD (CS-GP-CUR) was synthesized and evaluated as an antibacterial wound dressing [152]. CS-CG-CUR applied on rat cutaneous wounds infected with *S. aureus* reduced bacterial load and improved the healing process. In vitro assays showed that CS-CG-CUR had higher inhibition zones than CS-CG, antioxidant activity, and downregulation of the granulation tissue gene and proteins [152].

CUR incorporated in a transdermal patch of CS and polyvinyl alcohol (PVA/CS/CUR patch) produced inhibition zones of 14, 15, 18, and 20 mm against *B. subtilis*, *S. aureus*, *E. coli*, and *P. aeruginosa*, respectively (values of 11, 12.5, 15, and 14 mm, respectively, were shown in Table 2 of the original manuscript) [153]. These values were higher than those observed for the control patch without CUR (PVA/CS, 9–13 mm). The healing effect of the PVA/CS/CUR patch was evaluated in vivo in surgical wounds created on the dorsal surface of albino Wistar rats. The wounds were treated with the PVA/CS/CUR patch, which was replaced every four days and observed for 16 days. The healing process (scab formation, wound contraction, reduction in inflammation, collagen arrangement and deposition, and hair growth) observed in the rats treated with PVA/CS/CUR patch was faster than that observed in rats either treated with PVA/CS patch or untreated (control) [153].

Crude CUR extracted from turmeric was cross-linked with CS and PVA membranes [154]. The agar diffusion method showed a concentration-dependent effect of CUR mixed with CS-PVA against *Pasteurella multocida*, *S. aureus*, *E. coli*, and *B. subtilis*. The inhibition zones obtained for CUR with CS-PVA were higher than those observed for CUR alone and CS-PVA without CUR for all species evaluated. The total phenolic and flavonoid contents and the scavenging activity increased in samples of CUR with CS and PVA. The combination of CUR with CS and PVA applied twice a day in surgical wounds performed on the dorsal region of rabbits resulted in increased healing and wound reduction after 14 days [154].

The MIC of CUR encapsulated in CS nanoparticles (400 μg/mL) was higher than free CUR (200 μg/mL) against both *S. aureus* and *C. albicans* [155]. Confocal microscopic images showed the penetration of CS nanoparticles into the polymicrobial biofilm formed by both species on silicone surfaces. CUR in CS inhibited the formation of mono- and dual-species biofilms in a concentration-dependent manner and free CUR was slightly more effective. Conversely, CUR encapsulated in CS resulted in a higher reduction in preformed and mono- and dual-species biofilms than free CUR. Biofilms treated with CUR in CS nanoparticles showed reduced thickness and increased dead cells [155].

The in silico analysis showed the binding affinity of CUR in CS nanocomposite to NS5B polymerase of hepatitis C virus genotype 4A (HCV-4a), suggesting an inhibitory effect against virus replication [156]. Subsequently, the in vitro assay showed that the non-toxic concentration of CUR in CS inhibited the viral entry into Huh7 cells (derived from human hepatoma cells). The Western blot assay showed decreased expression of the HCV core protein of HCV-infected Huh7 cells treated with CUR in CS, indicating an antiviral activity of the nanocomposite against viral replication and entry [156].

Nanoparticles of CS and milk proteins were loaded with CUR and sprayed on potato plants infected with the potato virus Y (PVY) [157]. The ELISA assay showed higher antiviral activity of CUR in nanoparticles of CS and milk proteins than in the native form. The antiviral effect was concentration dependent. CUR-loaded nanoparticles also increased the chlorophyll content and the activity of antioxidant enzymes (peroxidase and polyphenol oxidase) of the infected plants [157].

### 3.7. CUR in Other Polymeric DDS

Antimicrobial studies with CUR loaded in other polymeric DDSs are summarized in Table 7.

Formulations of alginate foam loaded with CUR in PEG 400, γ-CD, and PEG + β-CD were synthesized as topical PSs to be used in aPDT for infected wounds [158]. In vitro experiments showed that all formulations resulted in complete photokilling of *E. faecalis* after 9.7 J/cm^2^ light irradiation. Conversely, only foam with CUR in PEG 400 reduced by 81% the survival of *E. coli* under irradiation of 29 J/cm^2^ [158].

A wet-milling technique was used to obtain nanoparticles of CUR (CUR-NP) without any polymer [159]. The antimicrobial effect of CUR-NPs was investigated against Gram-positive bacteria (*S. aureus* and *B. subtilis*), Gram-negative bacteria (*E. coli* and *P. aeruginosa*), and fungi (*Penicillium notatum* and *A. niger*) by agar dilution and well-diffusion methods. The MIC values of CUR-NPs were lower than those of CUR for all species evaluated, except for *P. notatum*, whose growth was not inhibited by any formulation. Bacteria showed higher susceptibility to CUR-NPs than fungi. The inhibition zones found for CUR-NPs against all bacteria were higher than those observed for CUR. CUR-NPs were more effective against Gram-positive than Gram-negative species. TEM images showed CUR-NPs anchored to the cell wall of *S. aureus* followed by cell disruption [159]. The same CUR-NPs showed lower Minimal Bactericidal Concentration (MBC) than CUR against *Micrococcus luteus*, *S. aureus*, *E. coli*, and *P. aeruginosa* [160]. The antibacterial effect of CUR-NPs was concentration dependent. Moreover, CUR-NPs were more toxic against cancer cells than mammalian cells [160].

A high-throughput screening showed that the amino acid proline, polyphenol tannic acid, and nitrogen-containing polymers polyquaternium-10 and polyvinylpyrrolidone (PVP) were able to stabilize CUR in aqueous NPs [161]. The MICs of CUR in the NP against *E. coli* were 400 μM for the stabilizer prolines, polyquaternium-10 and PVP, and 500 μM for tannic acid [161].

The antibacterial activity of CUR in NPs of different surfactants against *L. monocytogenes* showed the highest inhibitory effect of CUR in NPs of cetyltrimethylammonium bromide (CTAB) than CUR in NPs of Tween 20 and sodium dodecyl sulfate [162].

CUR encapsulated in NPs of polylactic acid and dextran sulfate resulted in the photoinactivation of MRSA, *S. mutans*, and *C. albicans* [163]. The incorporation of CTAB produced cationic NPs, which were toxic against microbial and mammalian cells. Mono- and mixed-species biofilms had lower susceptibility to aPDT than planktonic cultures, and free CUR resulted in higher photoinactivation of biofilms than CUR in NPs [163]. The aPDT mediated by free CUR and CUR in cationic NPs reduced the *C. albicans* load from the tongues of mice with induced oral candidiasis [164]. Conversely, aPDT mediated by CUR in anionic NPs did not reduce the fungal load in vivo [164].

CUR-NPs were synthesized by sonication without polymer and evaluated against multidrug-resistant *P. aeruginosa* strains isolated from burn wound infections [165]. The MIC of CUR-NPs was 128 μg/mL, while the MIC of CUR was 256 μg/mL. CUR-NPs inhibited the biofilm formation of bacterial strains and reduced preformed biofilms by 46%. CUR-NPs also downregulated the expression of virulence genes involved in biofilm formation (*rsmZ* and *lecA*) and efflux pumps (*mexD*, *mexB*, and *mexT*) of *P. aeruginosa* [165].

An inhalable formulation of nano-in-microparticles (NIMs) was synthesized by incorporating CUR in NPs of PLGA, which was embedded in a mannitol matrix by spray drying [166]. The aPDT mediated by NiMs reduced *Staphylococcus saprophyticus* subsp. *bovis* and *E. coli* DH5 alpha by 6.1 and 1.6 log_10_, respectively. TEM images showed NiMs surrounding the bacterial cells [166].

Free CUR presented a higher antibacterial effect than CUR-loaded nanocapsules of Eudragit L-100 against *L. monocytogenes* in gerbils [167]. However, animals treated with CUR in nanocapsules showed reduced hepatic injury, evaluated by increased antioxidant capacity, pyruvate kinase, and Na^+^/K^+^ATPase activities and decreased lipoperoxidation [167].

Nanocurcumin (nCUR) was added to discs of Activa BioActive Base/Liner (ABBL), which is a pulp cap agent used to treat dental pulp exposure [168]. ABBL discs with nCUR showed antibacterial activity against *S. mutans*, with inhibition zones up to 23 mm. After aging for up to 60 days, the discs were used as a substratum for biofilm formation and submitted to light irradiation. The aPDT reduced the viability of *S. mutans* biofilm proportionally to the nCUR concentration and the highest reduction was observed for discs submitted to aging for 60 days [168].

The PS indocyanine green was conjugated to nCUR and metformin, which is a drug used to treat type 2 diabetes and as a conjugated PS in aPDT against biofilms of *E. faecalis* formed in dental root canals for 14 days [169]. For aPDT, two light sources were used: a diode laser (810 nm) for indocyanine green and a LED (450 nm) for nCUR. The aPDT mediated by the conjugated PS followed by irradiation with both light sources (laser followed by LED and LED followed by laser) reduced biofilm viability by 82.74–83.84%. SEM images showed the surface of root canals with reduced bacterial cells for samples submitted to aPDT [169].

A solution of polyvinyl acetate and CUR (PVAc-CUR) was prepared and used to coat the surface of PET and polyvinylidene chloride (PVDC) film [170]. Samples of the coated films inoculated with *S. aureus* or *S. typhimurium* were irradiated and bacterial killing was improved with increasing CUR concentrations and light doses. SEM images showed the disruption of *S. aureus* membrane after aPDT and TEM images of *S. typhimurium* submitted to aPDT showed flagella disappearance and morphological changes. The antibacterial effect of PVAc-CUR against *S. aureus* was stable after 30 days and there was no significant reduction in antibacterial activity after multiple challenges with the bacterium. Samples of pork meat packaged with PVDC films with PVAc-CUR coating and submitted to irradiation showed a reduction in total viable counts and inhibition of meat degradation during storage [170].

CUR in NPs of PLGA was synthesized and incorporated in modified orthodontic adhesives (MOA) [171]. MOA was used to bond orthodontic brackets to the enamel surface of extracted human teeth. The shear bond strength of MOA reduced as the concentration of CUR in NPs increased. Samples of brackets bonded to enamel with MOA were submitted to aging up to 180 days. Biofilms of *S. mutans* were developed on aged samples, which were then irradiated. MOA reduced the biofilm biomass in non-irradiated samples submitted to aging up to 60 days. The aPDT reduced the biofilm biomass by up to 94.1% in samples aged up to 120 days [171].

Cellulose sponges loaded with CS and inclusion complex of CUR in β-CD was synthesized for the treatment of chronic wounds and evaluated in vitro [172]. The sponges reduced the viability of *S. aureus* and *E. coli* by 99.89% and 99.99%, respectively. Moreover, the sponges did not show indirect toxicity against mammalian cells, which attached and proliferated into the sponges [172].

### 3.8. CUR with Metallic Nanoparticles

Metal complexation plays an important role in the therapeutic properties of CUR. The β-diketone moiety in the CUR chemical structure enables it to form complexes with metal ions [173]. A previous review summarized the antimicrobial activity of CUR and curcuminoid complexes with metals, such as boron, Ca^2+^, Cd^2+^, Cr^3+^, Co^2+^, Cu^2+^, Fe^3+^, Ga^3+^, Hg^2+^, In^3+^, Mn^2+^, Ni^2+^, Pd^2+^, Sn^2+^, Y^3+^, and Zn^2+^ against viruses, bacteria, and fungi [173]. Metals have also been combined with polymers to improve the biological effects of CUR and to be used as films, hydrogels, dressings, and other pharmaceutical formulations [174,175]. In this context, silver NPs (AgNPs) have been extensively used due to their antimicrobial activity (Figure 6) [176]. Antimicrobial studies with CUR complexes with metals are summarized in Table 8.

CUR-AgNPs were synthesized and evaluated against Gram-positive bacteria (*B. subtilis* and *S. aureus*) and Gram-negative bacteria (*E. coli* and *P. aeruginosa*) [177]. The MICs and MBCs of CUR-AgNPs (2.5–10 and 5–20 mg/mL, respectively) were lower than the MIC of CUR alone and AgNPs for all species, except for *P. aeruginosa*. The microbial susceptibility to CUR-AgNPs was *B. subtilis* > *S. aureus* = *E. coli* > *P. aeruginosa*. The time–kill curves showed a reduction of 3 log after 2 h for *S. aureus* and *E. coli* and complete growth inhibition after 6 h for all species. CUR-AgNPs at 10 x MIC suppressed the growth of *S. aureus*, *B. subtilis*, and *P. aeruginosa* for 2.4, 3, and 5 h, respectively, while *E. coli* showed no growth. The crystal violet assay showed that CUR-AgNPs at MBCs inhibited the biofilm formation of *S. aureus*, *E. coli*, *B. subtilis*, and *P. aeruginosa* by 78%, 82%, 85%, and 85%, respectively, which was confirmed by fluorescence microscopy and SEM [177].

AgNPs were synthesized using CUR as a reducing agent and stabilized by capping with PVP [178]. The disc diffusion test showed that CUR-conjugated AgNPs resulted in inhibition zones similar to those verified for penicillin against *S. aureus* and *Salmonella* spp. and greater than that of antibiotic against *Fusarium* spp. Conversely, there was no inhibitory effect for *E. coli*. CUR-conjugated AgNPs showed smaller inhibition zones than those observed for amoxicillin against *S. aureus*, *Salmonella* spp., and *E. coli*, and similar inhibition against *Fusarium* spp. Additionally, CUR-conjugated AgNPs in the presence of Al^3+^ were able to estimate the concentration of nucleic acid (DNA and RNA) using Resonance Rayleigh Scattering [178].

CUR-AgNPs prepared with PVP resulted in the complete inhibition of *S. aureus* biofilm formation at concentrations of 30 μg/mL of CUR and 3.75 μg/mL of Ag [179]. The biofilm formation of *P. aeruginosa* was also completely inhibited by CUR-AgNPs with 40 μg/mL CUR and 5 μg/mL Ag. At higher concentrations (400 μg/mL CUR and 50 μg/mL Ag), CUR-AgNPs detached preformed biofilms of either species by nearly 70%, which was confirmed by SEM and confocal microscopy [179].

Nanoconjugates of CUR-Ag were prepared with PEG, which was used as a stabilizing agent [180]. The antibacterial effect of the conjugate, as well as the CUR nanoparticles (CUR-NP) and AgNPs, was tested by turbidimetric assay against *E. coli* XL-1. The results showed improvement of the antibacterial effect of the nanoconjugate, considering that 0.005 µM of the nanoconjugate was required to kill 80% of the bacterium, while 0.5 µM of CUR-NPs and 7 µM of AgNPs resulted in 80% and 60% of bacterial killing, respectively [180].

CUR was used as a reducing and capping agent to produce CUR-AgNPs, which showed a concentration-dependent effect on the death rate of *E. coli* and *B. subtilis* [181]. The MIC of CUR-AgNPs was 32 μg/mL for both bacteria. CUR alone combined with AgNPs stabilized by PVP showed synergism against both species. The growth kinetics showed complete inhibition of bacterial growth caused by CUR-AgNPs. Microscopy fluorescence and SEM and TEM images showed that CUR-AgNPs damaged the cell membrane. CUR-AgNPs also increased the intracellular ROS and the presence of antioxidants reduced the bacterial death caused by CUR-AgNPs [181].

Complexes of CUR with ruthenium(II) were synthesized and evaluated against ESKAPE pathogens (group of bacteria able to develop resistance and escape antibiotics, consisting of *E. coli*, *S. aureus*, *K. pneumoniae*, *A. baumannii*, *P. aeruginosa*, and *Enterococcus* sp.) [182]. Two complexes [Ru(NN)_2_(CUR)](PF_6_) [NN = bpy (1), phen (2)] were effective only against *S. aureus* (MIC of 1 μg/mL) but not against the other species (MIC from 8 to ≥64 μg/mL). The complexes were then evaluated against clinical isolates of MRSA and VRSA and showed MICs of 1–2 μg/mL. Time–kill kinetics showed that complex 1 at 10× MIC reduced the viability of *S. aureus* by 8.5 log in 24 h. The checkerboard assay presented synergism between complex 1 and antibiotics (meropenem, vancomycin, rifampicin, and minocycline) against *S. aureus*. Complex 1 at 10× MIC also reduced the biofilm biomass of *S. aureus* by 48%, which was higher than the reduction observed for the antibiotics vancomycin and levofloxacin. Complex 1 also decreased bacterial load by 0.35 log_10_ from a murine neutropenic thigh infection model, which result was similar to that observed for vancomycin (reduction of 0.48 log_10_) [182].

CUR encapsulated in sodium carboxymethyl cellulose silver nanocomposite films showed antibacterial activity against *E. coli*, evaluated by the disc diffusion method and kinetic growth [183].

Collagen scaffolds cross-linked (CSCL) with 20 and 10 μM CUR-caged AgNPs (CUR-AgNP) showed an antibacterial effect against *B. subtilis* and *E. coli* in a concentration-dependent manner, evaluated by the agar diffusion method [184]. The broth macro dilution assay also showed greater bacterial growth inhibition for CSCL with 20 μM CUR-AgNPs (95% and 80% for *E. coli* and *B. subtilis*, respectively) than for CSCL with 10 μM CUR-AgNPs (90% and 65% for *B. subtilis* and *E. coli*, respectively) [184].

Nanosilver nanohydrogels (nSnH) were synthesized by the polymerization of methacrylic acid as water/oil nanoemulsion, followed by the reduction of silver ions and cross-linking by gamma radiation [185]. A gel system of PVA/polyethylene oxide/carboxymethyl cellulose was blended with nSnH, *Aloe vera*, and CUR. The blend was coated on polyester fabric and evaluated as antimicrobial dressings. The antimicrobial activity of the dressings against *S. aureus* and *E. coli* was proportional to the concentration of nSnH. The dressings with nSnH and *A. vera* resulted in the complete eradication of *S. aureus*, while the dressing with nSnH and CUR reduced bacterial viability by up to 80%. A bacterial reduction of 92% was observed for dressings with nSnH, *A. vera*, and CUR. The antimicrobial dressings increased wound healing on mice by 100%, 80%, and 95% for nSnH dressings with *A. vera*, CUR, and *A. vera* + CUR, respectively. The dressings with nSnH and *A. vera* showed more effective antimicrobial and healing effects than the dressings with CUR [185].

A nanocomposite of CUR and zinc oxide NPs showed MICs of 195 µg/mL for *S. epidermidis*, *Staphylococcus haemolyticus*, and 390 µg/mL for *S. saprophyticus* [186]. All strains were methicillin-resistant clinical isolates. The antibacterial activity of the nanocomposite was confirmed by time–kill curves and fluorescence microscopy. SEM images showed treated cells without structural integrity. Collagen membranes loaded with the nanocomposite showed antistaphylococcal activity by the agar diffusion assay [186].

Thermo-responsive hydrogels were synthesized with poloxamer 407 (P407), CUR solid dispersion (CUR-SD), and AgNPs obtained with sodium citrate (AgNP-citrate) [187]. Other formulations of P407+CUR-SD+AgNPs were synthesized with AgNPs stabilized with PVA or PVP. The MICs of the formulations were determined against *S. aureus*, *P. aeruginosa*, and *E coli*. The most effective formulation against the three species was P407+CUR-SD+AgNP citrate at a concentration of 25%. Among the three microorganisms, *S. aureus* was the least susceptible, while *E. coli* was the most susceptible species to the formulations [187].

CUR-AgNPs showed antifungal activity against fluconazole-resistant *Candida* spp. isolated from HIV patients [188]. Although the concentrations of CUR-AgNPs are not reported for the agar diffusion test, the largest inhibition zones were observed for *C. glabrata* (20.6 ± 0.8 mm) and *C. albicans* (20.1 ± 0.8 mm), while the smallest ones were verified for *C. tropicalis* (16.4 ± 0.7 mm). The antifungal activity of CUR-AgNPs was higher than free CUR and silver nitrate alone. The MICs of CUR-AgNPs were 31.2, 62.5, 250, 125, 125, and 62.5 µg/mL against *C. albicans*, *C. glabrata*, *C. tropicalis*, *C. parapsilosis*, *C. krusei*, and *C. kefyr*, respectively. Nonetheless, CUR-AgNPs were characterized only by UV-visible spectroscopy [188].

A bioactive composite consisting of gelatin, CUR, and AgNPs (Gel-CUR-Ag) showed antibacterial activity against *S. aureus* and *P. aeruginosa*, with increased inhibition zones observed for composites with decreased concentrations of gelatin [189]. However, the concentrations were reported as µL/mL. The MIC of Gel-CUR-Ag varied from 125 to 250 µL/mL for *S. aureus* and *P. aeruginosa* and the MBC of the composite was ≥250 µL/mL for the Gram-positive bacterium and 125–250 µL/mL for the Gram-negative species. Moreover, the composite also showed in vitro antioxidant activity and no cytotoxicity against mammalian cell lines [189].

Hybrid hydrogels (HGZ) were prepared using regenerated cellulose from sugarcane bagasse and zinc oxide NPs, which were synthesized from muskmelon seeds [190]. CUR was then loaded in the HGZ (HGZ-CUR) and the release study showed a higher release of CUR at a neutral pH (7.4) compared with an acidic pH (1.2 and 4). The disc diffusion assay showed antimicrobial activity of HGZ-CUR against *S. aureus* and *T. rubrum* [190]. A similar CS-based hydrogel was synthesized using ZnO NPs from muskmelon seeds and the cross-linker dialdehyde cellulose from sugarcane bagasse, and loaded with CUR (CHG-ZnO-CUR) [191]. The release of CUR was also higher at a pH of 7.4 than at an acidic pH. CHG-ZnO-CUR showed higher inhibition zones against *S. aureus* and *T. rubrum* than those observed for hydrogel without CUR and/or ZnO NPs [191].

A nanocomposite of highly cross-linked poly-CUR nanospheres and copper (II) oxide NPs reduced the viability of *E. faecalis* and *P. aeruginosa*, which was confirmed by the disc diffusion test [192]. The nanocomposite was more effective against *E. faecalis* than *P. aeruginosa* [192].

A hybrid inclusion complex consisting of AgNPs, CUR, and oxidized amylose (OA-Ag-CUR) at 2.5 mg/mL inhibited the growth of *S. aureus* and *P. aeruginosa* [193]. The antibacterial effect against the Gram-negative bacterium was higher than that observed against the Gram-positive one. Fluorescence microscopy images suggested that AO-Ag-CUR promoted damage to the bacterial cell membrane [193].

AgNPs were synthesized using the inclusion complex of CUR in hydroxypropyl-β-CD as a reducing and capping agent (CUR-AgNP) and loaded in bacterial cellulose (BC) hydrogels [194]. The disc diffusion assay showed that CUR-AgNP-loaded BC had mean inhibition zones of approximately 16, 10, and 11 mm against *P. aeruginosa*, *S. aureus*, and the emerging pathogenic yeast *Candida auris*. Moreover, the CUR-AgNPs showed antioxidant activity and the hydrogels showed cytocompatibility despite being hemolytic [194].

Cotton fabrics were starched and coated with zinc oxide NPs (ZnO-NPs) [195]. Starch was used to stabilize the ZnO-NPs on the cotton fabrics and resulted in increased content and reduced leaching of ZnO-NPs. The ZnO-NPs-coated cotton starch showed antimicrobial activity against *S. aureus* and *E. coli* (reductions of 96% and 76%, respectively). The ZnO-NPs-coated cotton fabrics were functionalized with CUR, which resulted in a 100% viability reduction in both species. A ZnO-Ag nanocomposite was prepared and coated on cotton fabrics, which reduced *S. aureus* and *E. coli* by 100% and 90.5%, respectively [195].

A pH-responsive delivery system of N-succinyl modified CS coated ZnO-NPs were synthesized and conjugated to CUR using 1,1′-carbonyldiimidazole to activate the carboxyl groups (CS-ZnO-CUR) [196]. After 16 h at pHs of 5.2 and 7.4, the release of CUR was 95% and 40%, respectively. The MICs values of CS-ZnO-CUR were 10 and 5 μg/mL against *S. aureus* and *E. coli*, corresponding to a reduction of 25- and 50-fold, respectively, compared with free CUR. CS-ZnO-CUR showed MBC values of 50 and 25 μg/mL against *S. aureus* and *E. coli*, respectively, which were 10- and 40-fold lower, respectively, than the values observed for free CUR. Compared with ZnO NPs, the MIC and MBC values of CS-ZnO-CUR reduced 25- and 20-fold, respectively, for both bacteria. A dose-dependent cytotoxic effect of CS-ZnO-CUR was observed against breast cancer cells but not against normal cells, and flow cytometry results showed an induction of late apoptosis in cancer cells treated with CS-ZnO-CUR [196].

Complexes of CUR and titanium dioxide (TiO_2_) NPs were loaded in CS and used to fabricate membranes of polypropylene (MCUT) [197]. Although CUR and TiO_2_ alone showed inhibition zones of 2 mm against *S. aureus* and *E. coli*, values of 4.8–8.1 mm were observed for the complexes. The MCUT samples resulted in mean inhibition zones of 6–8 mm for *E. coli* and of 9–11 mm for *S. aureus*. The MCUT samples were used as dressings on infected wounds made on rats and the results showed increased healing, wound contraction, reduction of 3.25 log_10_ of MRSA recovered from wounds, better epithelialization, collagen organization, and no inflammation after 14 days [197].

CUR isolated from *Curcuma pseudomontana* was used as a reducing and stabilizing agent in the synthesis of CUR-gold NPs (CUR-Au-NPs) [198]. The agar diffusion method showed that CUR-Au-NPs had a concentration-dependent antibacterial effect against Gram-positive and Gram-negative species. At 300 μg/mL, CUR-Au-NPs resulted in inhibition zones of 28, 26, 25, and 23 mm against *E. coli*, *B. subtilis*, *S. aureus*, and *P. aeruginosa*, respectively, while at 100 μg/mL, the inhibition zones were 12, 11, 11, and 13 mm, respectively. CUR-Au-NPs also showed in vitro anti-inflammatory and antioxidant activities [198].

### 3.9. CUR in Mesoporous Particles

Porous materials are structures with ordered pores ranging from nanometer to micrometers, which are classified as microporous (less 2 nm), mesoporous (from 2 up to 50 nm), and macroporous (above 50 nm) [199]. Porous materials can be synthesized using carbon, silica, and metal oxides [200]. Mesoporous silica nanoparticles (MSN, Figure 7) are inorganic scaffolds [201], which seemed ideal carriers for hydrophobic drugs due to their well-defined structure, large specific surface area, and versatile chemistry for functionalization [202]. The pore size and volume and the surface area, as well as the surface functionalization of the mesoporous material, determine the drug load and release [203]. Moreover, mesoporous materials can be modified or functionalized to control drug release under environmental stimuli, such as pH, temperature, or light. These stimuli-responsive DDS, or smart DDS, prevent undesirable drug release before reaching the target tissue (“zero premature release”) [203]. Antimicrobial studies with CUR in porous DDSs are summarized in Table 9.

A trio-hybrid nanocomposite of MSN containing copper, decorated with Ag nanoparticles and loaded with CUR showed the photokilling of *E. coli* [204]. The nanocomposite showed a positive surface charge and the presence of Ag increased ROS production. After irradiation, the trio-hybrids with CUR at 1.5 μM reduced bacterial viability by 5 and 4 log compared with free CUR and nanocomposite without CUR, respectively. At 3 μM of CUR, the trio-hybrids eradicated the bacterial cells. SEM images showed bacterial cell damage after nanocomposite-mediated aPDT [204].

A bionanocomposite film of CS and CUR-loaded MSN was synthesized and showed antibacterial activity against *S. aureus* and *E. coli*, measured by the inhibition zone [205]. The inhibition of *S. aureus* was higher than that observed for *E. coli*, although the antibacterial effect of CS with CUR was higher than the bionanocomposite film [205].

Another type of mesoporous material constituted by the iron–phenanthroline nanocomplex was used to encapsulate CUR (NCIP) and as an antiretroviral agent in human microglial cells infected with HIV-1 [206]. NCIP at 5 and 8 mg was not toxic to HIV-transfected microglial cells. Immunofluorescent staining showed that NCIP reduced the expression of HIV-p24 by 41% while free CUR accounted for 24% of reduction, suggesting suppression of HIV replication. NCIP also reduced the expression of nitric oxide, interleukin-8 (IL-8), and TNF-α by 50.2%, 41%, and 61.2%, respectively, suggesting an anti-inflammatory effect. This finding was confirmed by the flow cytometry assay, which showed a decrease in IL-8 and TNF-α expression by 72% and 56%, respectively. An antioxidative effect was also observed, considering that NCIP reduced the gene expression of heme oxygenase and increased the gene expression of catalase [206].

An asymmetric lollipop-like mesoporous silica NP with a head of spherical Fe_3_O_4_@SiO_2_ and a tail of ethane bridged periodic mesoporous organosilica (EPMO) nanorod was synthesized and showed hydrophobic/hydrophilic independent spaces, which were co-loaded with hydrophobic CUR and hydrophilic gentamicin sulfate (Fe_3_O_4_@SiO_2_-GS&EPMO-Cur) [207]. Fe_3_O_4_@SiO_2_-GS&EPMO-Cur presented the independent release of both drugs, which did not interact with each other. Without the drugs, the NPs did not show cytotoxicity against human breast cancer cells. Conversely, Fe_3_O_4_@SiO_2_-GS&EPMO-Cur inhibited cancer cells by 89.6%. Fe_3_O_4_@SiO_2_-GS&EPMO-Cur with CUR at 13 μg/mL also showed antibacterial activity against *S. aureus* and *E. coli*, inhibiting their growth by 92.1% and 90%, respectively, which was measured by the absorbance of bacterial suspensions. Fe_3_O_4_@SiO_2_-GS&EPMO-Cur showed long-term antibacterial activity, inhibiting the growth of both bacteria after 60 h [207].

The Santa Barbara Amorphous-15 (SBA-15), which is an MSN with an ordered hexagonal channel structure, was coated on its inner and outer surface with melanin-like polydopamine (PDA), decorated in situ with Ag NPs, and loaded with CUR (CUR@SBA-15/PDA/Ag) [208]. The hemolytic activity of CUR@SBA-15/PDA/Ag on red blood cells was lower than that of the nanocomposite without CUR, SBA-15/PDA/Ag. CUR@SBA-15/PDA/Ag showed pH- and ROS-responsive release, considering the release of CUR was increased 12-fold at a pH of 4.5 and threefold in the presence of H_2_O_2_. The release of Ag was also increased at a low pH. CUR@SBA-15/PDA/Ag showed cytotoxicity against malignant cells and drug-resistant cancer cells and reduced the colony growth of *E. coli* and *S. aureus*. The antibacterial effect toward *E. coli* was more effective than *S. aureus* [208].

### 3.10. CUR in Graphene Nanocomposites

A graphene is a crystalline form of carbon arranged as a monolayer honeycomb lattice. Graphene types include graphene oxide, produced from the oxidation of graphite, and reduced graphene oxide, with application in biotechnology, biosensors, supercapacitors, and others [209]. Antimicrobial studies with CUR in composites of graphene are summarized in Table 10.

A nanocomposite of graphene-NH_2_, iron oxide NPs, and PEG (G-NH_2_–IONP–PEG) was synthesized and loaded with CUR [210]. The release of CUR from the nanocomposite was increased at a pH of 5 compared with the pH of 7.4. G-NH_2_–IONP–PEG reduced the colony count of *S. aureus* and *E. coli* in a concentration-dependent manner [210].

A cationic composite of CUR with reduced graphene oxide (CUR-rGO) showed minimum biofilm inhibitory concentration (MBIC) of 250 μg/mL against 4-week-old *E. faecalis* biofilms formed on the wells of a microtiter plate coated with dentin particles [211]. The aPDT mediated by CUR-rGO at lower concentrations than MBIC and LED light reduced the metabolic activity of the biofilm. SEM images showed a reduction in the aPDT-treated biofilm formed on dentin root canals of extracted human teeth. The aPDT also downregulated the genes involved in adherence, biofilm formation, and biofilm development of *E. faecalis* biofilms [211].

A nanocarrier of reduced graphene oxide and ZnO NPs (GrZnO) was synthesized and showed MICs of 125 and 250 μg/mL against two reference strains of MRSA, whereas the combination of CUR with GrZnO showed MICs of 31.25 and 62.5 μg/mL [212]. The combination also showed higher inhibition zones and reduction in bacterial viability than GrZnO, CUR alone, and vancomycin. CUR with GrZnO at sub-MIC suppressed the bacterial growth kinetics. The combination reduced the metabolic activity and biomass of preformed biofilms in a concentration-dependent manner. TEM images showed damage to the bacterial cell membrane treated with CUR + GrZnO. This result was confirmed by the protein leakage assay. Infected lesions on the skin of mice showed a reduction of 64% in the viability of MRSA and histopathological features similar to healthy tissue after treatment with CUR + GrZnO [212].

### 3.11. CUR in Quantum Dots

Quantum dots (QDs) are semiconductor particles at nanosize (up to 10 nm) with electrical and photoluminescence properties of biotechnological and biomedical applications, such as bioimaging and DDS [217]. Carbon dots are divided into carbon QD and graphene QD and are produced by top-down and bottom-up methods using bulk carbon material and molecular precursors, respectively [217]. Antimicrobial studies with CUR in QDs are summarized in Table 10.

CUR QDs (CQDs) were synthesized using a two-step wet milling method and showed a spherical shape and mean size of 2.5 nm [213]. CQDs presented antibacterial activity against *S. aureus*, MRSA, *E. faecalis*, *E. coli*, *K. pneumoniae*, and *P. aeruginosa* with a MIC range from 3.91 to 7.825 μg/mL, which was lower than the MIC of free CUR (175–350 μg/mL). When biofilms were evaluated, CQDs completely eradicated the biofilm formation of *E. coli* and inhibited the biofilm formation of MRSA, *S. aureus*, and *S. epidermidis* by 24.24–78.31%, whereas the biofilm of *P. aeruginosa* showed the lowest inhibition (11.94–39.55%). CQDs also reduced the biomass of preformed biofilms of *E. coli*, MRSA, *S. aureus*, and *S. epidermidis* by up to 100% and *P. aeruginosa* by up to 85.78%. Additionally, the interaction between CUR and biofilm matrix proteins was also verified [213].

CUR-derived carbon QDs (CUR-cQDs) were synthesized through simple one-step dry heating (pyrolysis) of CUR at temperatures from 120 to 210 °C [214]. CUR-cQD produced at 180 °C (CUR-cQD-180) showed the best antiviral activity against enterovirus 71 (EV71), which is responsible for illness in young children. CUR-cQDs reduced the apoptosis and plaque formation of human rhabdomyosarcoma (RB) cells infected with EV71. CUR-cQD-180 decreased the infectious viral titer by 85% and showed a MIC of 5 μg/mL, while the MIC of free CUR was >200 μg/mL. The immunoblotting assay showed that CUR-cQD-180 reduced the protein and proteases involved in EV71 replication. CUR-cQD-180 also showed antioxidant activity by scavenging ROS from infected cells. In vivo results showed that EV71-infected mice treated with CUR-cQD-180 had increased survival and decreased viral titers in brain and limb muscle tissues [214].

CUR and molybdenum disulfide QDs were hybridized (CUR-MQD) with a seed-mediated hydrothermal method [215]. The MIC of CUR-MQD ranged from <0.125 to 25 μg/mL against reference strains and clinical isolates of *K. pneumoniae*, *P. aeruginosa*, and *S. aureus*, the MBC ranged from 0.25 to 1 μg/mL and the MBIC ranged from <0.00625 to 0.125 μg/mL. CUR-MQD reduced the biofilm biomass of *K. pneumoniae* by 73–97%, which was confirmed by confocal microscopy. Spectrofluorimetry, fluorescence microscopy, and flow cytometry assays showed alterations in membrane lipids, membrane permeabilization, and depolarization of *K. pneumoniae* cells treated with CUR-MQDs, without ROS production. The oral administration of CUR-MQDs in rats did not change the serum, hematological profile, and histopathological features of the liver and kidney [215].

CUR was added to graphene QDs (CUR-GQDs), which was used as a PS in aPDT against three periodontal pathogens (*A. actinomycetemcomitans*, *P. gingivalis*, and *P. intermedia*) [216]. The aPDT mediated by GQDs alone, free CUR, and CUR-GQDs reduced the viability of a mixed bacterial culture by 73.1%, 82.2%, and 93%, respectively, and the biomass of mixed-species biofilm by 56.4%, 61.3%, and 76%, respectively. After aPDT, the multispecies biofilms also showed a downregulation of genes related to biofilm formation. Moreover, CUR-GQDs showed low toxicity to human gingival fibroblast cells and aPDT resulted in ROS production from a mixed bacterial culture [216].

### 3.12. CUR in Films, Hydrogels, and Other Nanomaterials

Antimicrobial studies with CUR in films, hydrogels, and other nanomaterials are summarized in Table 11.

CUR in silica NPs (CUR-SiNPs) were synthesized and used as a PS in aPDT with a light fluence of 20 J/cm^2^ [218]. *S. aureus* and *P. aeruginosa* treated with CUR-SiNPs at 1 mg/mL showed reductions of 1.2 and 1 log_10_(CFU/mL), respectively, while aPDT resulted in a reduction of >6 log_10_ for both species. Biofilms submitted to aPDT mediated by CUR-SiNPs at 50 μg/mL and 1 mg/mL showed reductions of >1 and >3 log_10_, respectively, for *S. aureus* and >1 and >5 log_10_, respectively, for *P. aeruginosa* [218].

CUR was loaded in halloysite nanotubes (HNT), which are tubular aluminosilicates made from kaolin, and the nanoformulation was coated with dextrin (DX), which worked as stoppers, slowing the CUR release [219]. Before the in vivo evaluation of the nanoformulation CUR-HNT-DX on the nematode *Caenorhabditis elegans*, in vitro assays showed higher antibacterial activity of the nanoformulation against the nematode pathogen *Serratia marcescens* than that against the nematode food *E. coli*. CUR-HNT-DX also suppressed the virulence factor of *S. marcescens*, such as swarming motility, synthesis of prodigiosin (a quorum-sensing molecule), and biofilm formation. *E. elegans* fed with *E. coli* and the nanoformulation showed increased body development, while *S. marcescens* reduced worm body length. CUR-HNT-DX increased nematode reproduction, which was slightly lower in *S. marcescens*-fed worms. *S. marcescens*-fed nematodes showed reduced longevity, whereas CUR-HNT-DX treatment increased worm longevity [219].

Exosomes isolated from cultured cells were used as DDSs and loaded with CUR [220]. The exosomes were engineered to express on its surface a single-chain variable fragment of an HIV-1 envelop protein-specific antibody (10E8). Confocal microscopy and flow cytometry showed that 10E8-exosomes bound to cells expressing the HIV envelop (Env^+^). CUR-loaded 10E8-exosomes reduced the viability of Env^+^ cells, suppressed HIV infection and replication, and induced death in cells infected with HIV-1. In vivo results showed that 10E8-exosomes injected in mice were distributed at the tumor sites, liver, and intestinal tracts, and CUR-loaded 10E8-exosomes suppressed tumor growth [220].

Electrospun CUR nanofibers were synthesized and evaluated against a 7-day-old biofilm of *Actinomyces naeslundii* [221]. The highest reduction in bacterial viability was observed for CUR irrigants at 2.5 and 5 mg/mL, with or without light irradiation (1200 mW/cm^2^ for 4 min at 30 s intervals). CUR nanofibers decreased bacterial viability only when irradiated [221].

A co-delivery nanofiber (NF) consisting of CUR-loaded β-CD-grafted graphene oxide core (CUR@CD-GO) and gallic acid-loaded CS shell (Ga@CS) was synthesized (CUR-Ga NF) [222]. The agar diffusion test showed that NFs of CUR-Ga and CUR@CD-GO produced inhibition zones of 21.6 and 18 mm against *B. cereus*, respectively, and 16.4 and 14.6 mm against *E. coli*, respectively. The MIC of CUR-Ga and CUR@CD-GO NFs were 42.6 and 54.5 µg/mL, respectively, against *B. cereus*, and 51.3 and 63.25 µg/mL, respectively, against *E. coli*. Moreover, the CUR-Ga NFs showed in vitro antioxidant activity, toxicity to cancer cells, and in vivo anti-inflammatory effect in mice [222].

A multinanofiber composite film was obtained by the hybridization of nanofibrillated bacterial cellulose and chitin nanofibers and in situ precipitation of CUR micro/nanoparticles [223]. The size of CUR particles was proportional to the concentration of the CUR solution used in the precipitation process. The films inhibited the viability of *S. aureus* and *E. coli*, which was confirmed by SEM and confocal microscopy. The films also showed a color change from yellow to reddish at an increased pH and in the presence of boric acid, which is harmful to human health [223].

Nanofiber scaffolds of polyurethane and cellulose acetate were prepared with CUR and reduced graphene oxide decorated with AgNPs [224]. The scaffolds presented higher antibacterial activity against *Pseudomonas* sp. compared with *S. aureus*. In vivo assays showed that the scaffolds were biocompatible in mice and increased wound healing in rats [224].

A hybrid nanofibrous scaffold was synthesized by electrospinning poly(_L_-lactic acid) and poly(citrate siloxane), adding CUR and depositing dopamine on the matrix (PPCP) [225]. The scaffold showed photothermal properties under near infrared (NIR) irradiation, which released CUR. PPCP also showed antioxidant activity, compatibility to blood and skin cells, and antibacterial activity against *S. aureus* and *E. coli*. In vitro and in vivo analyses showed the inhibition of tumor growth after NIR irradiation of PPCP. PPCP accelerated wound healing and reduced the growth of *S. aureus* from the infected wound tissue [225].

A stretchable nanofibrous mat composed of segmented polyurethane (SPU) based on di-block (PCE) and tri-block (PECE) PCL and PEG was fabricated and loaded with CUR [226]. The water uptake assay showed that PECE was more hydrophilic than PCE and SPU. The PEG content increased the mean diameter of nanofibers. The incorporation of CUR also increased the mean diameter of the nanofiber, its viscosity, conductivity, and surface tension and decreased its porosity. The scaffolds showed improved mechanical properties (elongation at break and tensile strength). PECE also showed faster biodegradation than that observed for PCE and SPU and maximum CUR release (85%). All the mats with 10% loaded CUR showed higher growth inhibition of *S. aureus* and *E. coli* than the mats with 5% loaded CUR [226].

CUR solid dispersion (CSD) was prepared with PVP and loaded in temperature-sensitive in situ hydrogels (CSDG) [227]. CSDG formed gel in a simulated vaginal fluid. CSD and CSDG improved CUR release compared with CUR powder and CUR in hydrogel. Although CSDG showed, in vitro, weak or no antimicrobial effect against *S. aureus* and *E. coli*, respectively, the in vivo evaluation of CSDG on an injured rat vaginal infection model showed a reduction in the viability of *S. aureus* and *E. coli*. The in vivo antibacterial effect was ascribed to the immune response and the presence of *Lactobacillus* in the local environment of the vagina. Moreover, CSDG also improved vaginal inflammation and wound healing [227].

A gelatin-based film was prepared with CUR and the surfactant sodium dodecyl sulfate [228]. The CUR–gelatin film blocked UV light and showed improved mechanical and water vapor barrier properties. The CUR–gelatin film showed antioxidant activity and a higher reduction in the viability of *E. coli* than *L. monocytogenes* [228]. Similar antibacterial results were observed for films of carboxymethyl cellulose with CUR and ZnO [229]. However, a pectin-based film with CUR and sulfur NPs reduced the viability of both species and the antibacterial activity was proportional to the concentration of sulfur NPs [230]. The pectin/CUR/sulfur NPs films showed a color change from yellow to reddish-brown at an increased pH from 2 to 12. The color change was also observed when the films were exposed to ammonia vapor and when used as shrimp packaging [230].

An edible film was synthesized by the dispersion of orange oil and CUR in guar gum, soy lecithin, and glycerol [231]. The moisture content of the films decreased as orange oil content increased. CUR and orange oil decreased the dissolvability and wettability of the film. The agar diffusion test showed that films with orange oil produced a higher inhibition zone against *E. coli* than against *B. subtilis*. Conversely, films with CUR and orange oil showed a higher inhibition zone against *B. subtilis* than against *E. coli*. The films also prevented mold formation on strawberries after seven days. Orange oil decreased water vapor permeability, which is desirable for edible films, while CUR increased water vapor permeability. Moreover, orange oil decreased the tensile strength of the film, and CUR did not affect its mechanical properties [231].

## 4. Conclusions and Future Perspectives

CUR has a broad-spectrum antimicrobial activity against viruses, bacteria, and fungi, including resistant and emergent pathogens. However, some species such as Gram-negative bacteria are less susceptible to CUR and aPDT. For those, the combination of CUR with antibiotics has been suggested, especially for antibiotic-resistant strains [228]. CUR showed synergism with polymyxin and protection against the side effects of polymyxin treatment, nephrotoxicity, and neurotoxicity [232]. Nonetheless, the evaluation of synergism requires accurate methods to study drug interaction, considering potential differences between the dose–response relationship of individual drugs and avoiding over- or under-estimation of interactions. For example, while the time–kill curve of *C. jejuni* treated with both cinnamon oil and ZnO NPs resulted in the over-estimation of synergism between the antimicrobials, the fractional inhibitory concentration index (FICI) method showed no synergism but only an additive effect [233]. The FICI method was not able to detect the synergism between binary combinations of antimicrobials (cinnamon oil, ZnO NPs, and CUR encapsulated in starch) at sub-MIC, which resulted in the non-turbidity of *C. jejuni*. In turn, mathematical modeling using isobolograms and median-effect curves showed synergism when CUR in starch was combined with other antimicrobials against *C. jejuni*, with bacterial reductions of 3 log for the binary combination and over 8 log for the tertiary combination. The mathematical modeling suggested that CUR in starch was the main antimicrobial responsible for the synergistic interaction [233].

In addition to the antimicrobial evaluation, in vitro and in vivo studies have demonstrated the cytocompatibility and biocompatibility of CUR in DDSs [109,116,117,118,121,124,148,149,156,160,172,189,196,206,216], suggesting that CUR-loaded DDSs might be safe. Although a plethora of DDSs has been developed to circumvent the hydrophobicity, instability in solution, and low bioavailability of CUR, several studies are still performed with free CUR dissolved in organic solvents [19,20,21,22,23,24,25,26,27,28,29,30,31,32,33,34,35,36,37,38,39,40,41,42,43,44,45,46,47,48,49,50,51,52,53,54,55,56,57,58,59,60,61,62,63,64,65,66,67,68,69,70,71,72,85,86,87,88,89,90]. Furthermore, compared to several in vitro investigations, few in vivo studies using animal infection models and scarce clinical trials have been reported. A randomized clinical trial showed that aPDT mediated by free CUR improved gingivitis in adolescents under fixed orthodontic treatment but did not reduce dental plaque accumulation after 1 month [234]. Clinical improvements after CUR-mediated aPDT were also observed for periodontal diseases, although few studies have evaluated the microbiological parameters [63,89]. Therefore, the improvement of clinical parameters might be due to the anti-inflammatory effect of CUR/aPDT instead of their in vivo antimicrobial activity. Nonetheless, randomized clinical trials evaluating CUR in DDSs against infections are required.

As a note on the future use of CUR, the incorporation of CUR in DDS and other pharmaceutical formulations allows its clinical use especially as an adjuvant agent to conventional antimicrobial agents. Such a combination can be an important weapon in the battle against resistant strains and emergent pathogens. The use of stimuli-responsive (or smart) DDS can also improve CUR delivery and its therapeutic effect on the target tissue. The combination of polymeric and metallic carriers may also enhance the therapeutic activity of CUR. Nonetheless, the degradation of DDS and its clearance from the body are other issues that require further investigation [18]. The evidence produced so far about the antimicrobial activity of CUR in DDSs supports future in vivo and clinical studies, which may pave the way for industrial production.

## Figures and Tables

**Figure 1 ijms-22-07130-f001:**
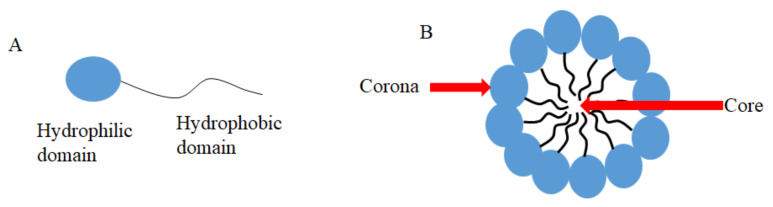
Schematic representation of: (**A**) an amphiphilic molecule and (**B**) an assembled micelle.

**Figure 2 ijms-22-07130-f002:**
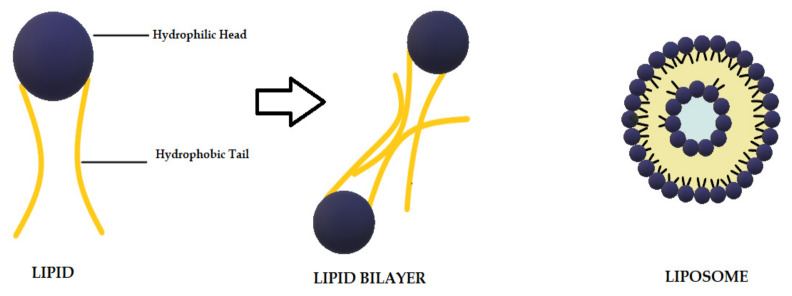
Schematic representation of the liposome structure.

**Figure 3 ijms-22-07130-f003:**
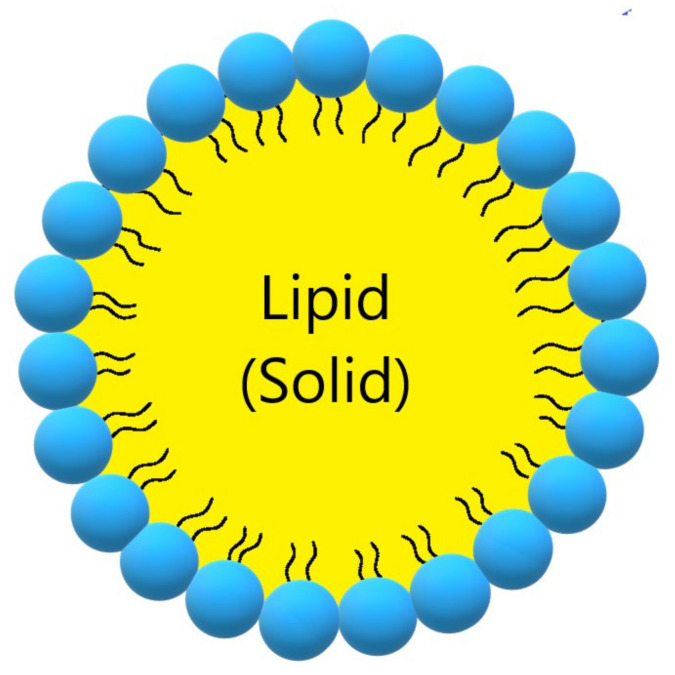
Schematic representation of solid lipid nanoparticle.

**Figure 4 ijms-22-07130-f004:**
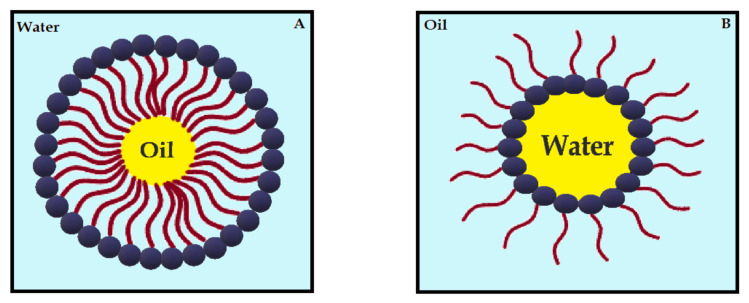
Schematic diagram of oil-in-water nanoemulsion (**A**) and water-in-oil nanoemulsion (**B**), stabilized by surfactants.

**Figure 5 ijms-22-07130-f005:**
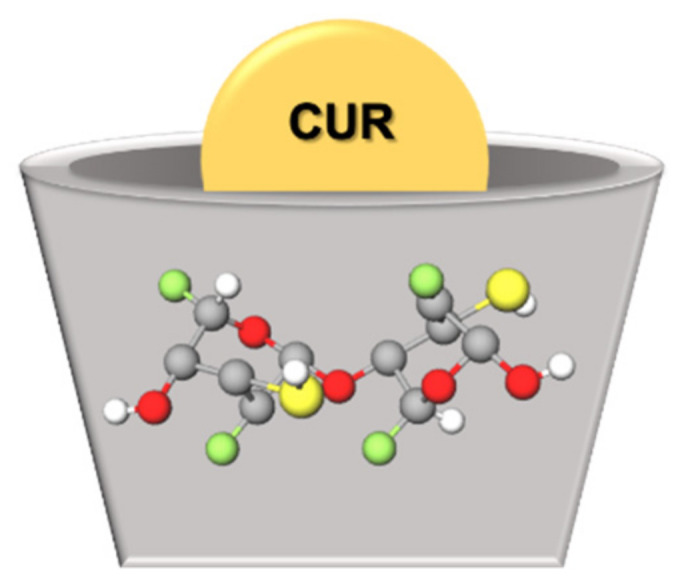
Schematic representation of CUR in CD.

**Figure 6 ijms-22-07130-f006:**
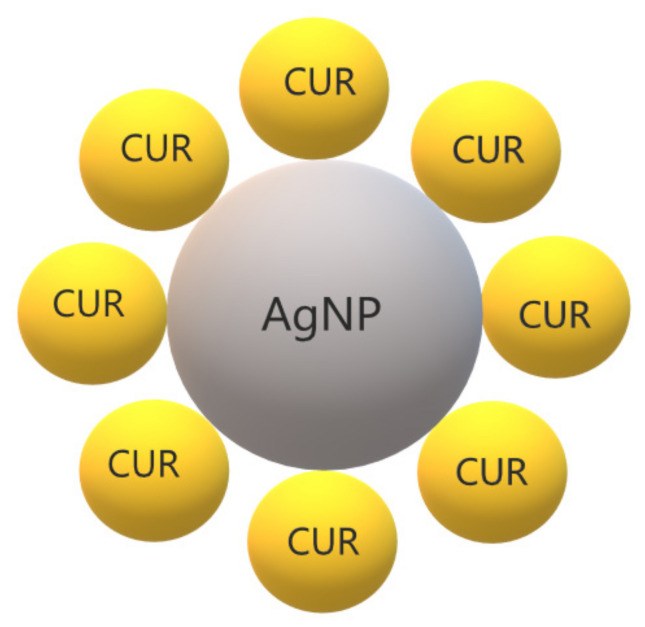
Schematic representation of CUR in silver nanoparticles.

**Figure 7 ijms-22-07130-f007:**
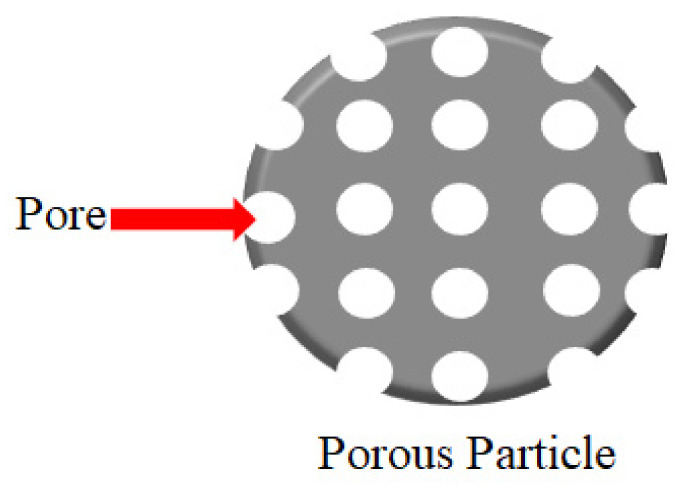
Schematic representation of a porous particle.

**Table 1 ijms-22-07130-t001:** In vitro and in vivo studies using free CUR and curcuminoids as antimicrobial.

Solvent	Microorganism	Culture	Antimicrobial Method	CUR Concentration	Light/Ultrasonic Parameters	Reference
DMSO (0.4%)	ZIKV	Cell infection	IC_50_	5.62–16.57 µM	-	[19]
	>DGEV		>IC_90_			
N/R	HPVA	Cell infection	Viral survival	0.015 mg/mL	-	[20]
	Tulane V					
N/R	KSPV	Infected cells	EC_50_	Up to 6.68 µM	-	[21]
Aqueous *Piper nigrum* seed extract	SARS-CoV-2	Cell infection	IC_50_ Plaque reduction	0.4 µg/mL	-	[22]
DMSO (<0.4%)	SARS-CoV-1	Cell infection	Inhibiton of viral replication	20 µM	-	[23]
N/R	SARS-CoV	In vitro	Viral inhibition	23.5 µM	-	[24]
N/R	SARS-CoV	In vitro	papain-like inhibition	5.7 µM	-	[25]
DMSO (1 *w*/*v*)	*S*. *aureus*	Planktonic	Inhibition zone MIC	600 and	-	[26]
	*E. coli*			400 µg/mL		
DMSO	MRSA	Planktonic	MIC FICI	15.5 µg/mL	-	[27]
N/R	*S*. *aureus*	Planktonic	Colony count	100 µg/mL	8 or 20 J/cm^2^	[28]
	MSSA					
	MRSA					
DMSO (10%)	*S*. *aureus*	Biofilm	aPDT	20, 40, and 80 µM	5.28 J/cm^2^	[29]
DMSO	VRSA	Biofilm/animal infection model	MIC MBC	156.25 µg/mL	20 J/cm^2^	[30]
N/R	*S. aureus*	Animal infection model	aPDT	78 µg/mL	60 J/cm^2^	[31]
DMSO	*S. aureus*	Infected fruit	Survival fraction	100 nM	1.5 and 9 J/cm^2^	[32]
N/R	*S. aureus*	Planktonic	PDI	40 and 80 µM	15 J/cm^2^	[33]
	*E. coli*					
Tween 80 (0.5%)	*S. aureus*	Planktonic	CFU/mL	300 and 500 µM	0.03–0.05 W/cm^2^	[34]
N/R	*S. aureus*	Biofilm	Confocal microscope	N/R	170 µmol m^2^ s^1^	[35]
DMSO (0.5%)	*S. aureus*	Biofilm	SDT aPDT SPDT	80 µM	100 Hz 15 and 70 J/cm^2^ 100 Hz, 15 and 70 J/cm^2^	[36]
DMSO	*E. coli*	Planktonic	MIC Inhibition zone	110, 220 and 330 µg/mL	-	[37]
DMSO	*E. coli*	Planktonic	OD_600nm_	8,16, 32, and 64 µg/mL	-	[38]
N/R	*S. dysenteriae*	Planktonic	MIC/MBC	256 and	-	[39]
	*C. jejuni*			512 µg/mL		
Edible alcohol	*E. coli*	Planktonic	aPDT	5, 10, and 20 µM	3.6 J/cm^2^	[40]
DMSO	*H. pylori*	Planktonic biofilm	MIC MBC aPDT	50 µg/mL	10 mW/cm^2^	[41]
DMSO	*P. aeruginosa*	Biofilm	aPDT CFU/mL	N/R	5 and 10 J/cm^2^	[42]
DMSO	Imipenem-resistant *A. baumannii*	Planktonic	aPDT	25, 50, 100, and 200 µM	5.4 J/cm^2^	[43]
DMSO (2%)	*P. aeruginosa*, *A. baumannii*, *K. pneumoniae*, *E. coli*, *E. faecalis*	Planktonic	MIC/FICI	128-256 µg/mL	-	[44]
N/R	*C. difficile*, *C. sticklandii*, *B. fragilis*, *P. bryantii*	Planktonic	Viable cell number	10 µg/mL	-	[45]
N/R	*B. subtillis*, *E. coli*, *S. carnosus*, *M. smegmatis*	Planktonic	MIC/MBC	Up to 25 µM	-	[46]
N/R	MRSA	Planktonic/animal infection model	MIC	4–16 μg/mL	-	[47]
	MSSA			2–8 μg/mL		
	*E. coli*			8–32 μg/mL		
N/R	*E. faecalis*, *S. aureus*, *B. subtillis*, *P. aeruginosa*, *E. coli*	Planktonic	MIC	156 μg/mL	-	[48]
DMSO (0.5%)	*A. hydrophila*, *E. coli* *E. faecalis*, *K. pneumoniae*, *P. aeruginosa*, *S. aureus*, *C. albicans*	Planktonic	MIC/MBC/ FICI/aPDT	37.5–150 µg/mL	N/C	[49]
N/R	*E. faecalis*	Infection model	CFU/mL	1 µg/mL	-	[50]
Commercial solution	*E. faecalis*	Biofilm	aPDT	1.5 g/mL	20.1 J/cm^2^	[51]
Ethanol 99%	*A. hydrophila*, *V. parahaemolyticus*	Planktonic	aPDT/SDT	Up to 15 mg/L	N/C	[52]
DMSO (10%)	*E. faecalis*	Biofilm	MIC/MBC	120 mg/mL	-	[53]
N/R	*S. mutans*	Planktonic	aPDT	10 g/100cc	N/C	[54]
DMSO: ethyl alcohol	*S. mutans*, *S. pyogenes*	Planktonic	aPDT	3 mg/mL	28.8 J/cm^2^	[55]
DMSO (0.8%)	Caries isolated	Biofilm	aPDT	600 µg/mL	75 J/cm^2^	[56]
DMSO	*S. mutans*, *C. albicans*	Biofilm single/dual	MBEC	0.5 mM	-	[57]
DMSO (0.05 M)	*A. actinomycetemcomitans*	Planktonic	aPDT	40 µg/mL	300–420 J/cm^2^	[58]
DMSO (<1%)	*P. gingivalis*, *A. actinomycetemcomitans*	Planktonic	aPDT	20 µg/mL	6, 12 or 18 J/cm^2^	[59]
DMSO (0.5%)	*P. gingivalis*, *A. actinomycetemcomitans*, *C. rectus*, *E. corrodens*, *F. nucleatum*, *P. intermedia*, *P. micra*, *T. denticola*, *T. forsythis*	Biofilm	aPDT	100 mg/L	-	[60]
N/R	Subgingival plaque	Biofilm	aPDT	100 µg/mL	30 J/cm^2^	[61]
DMSO	*P. gingivalis*	Planktonic	MIC	12.5 µg/mL	-	[62]
Ethanol: DMSO (99.9%: 0.1%)	Periodontal pocket	-	aPDT	100 mg/mL	7.69 J/cm^2^	[63]
Tween 80	*Streptococcus* spp, *Staphylococcus* spp, Enterobacteriaceae, *C. albicans*	Clinical trail	aPDT	0.75 mg/mL	20.1 J/cm^2^	[64]
Sodium hydroxide: PBS	*C. albicans*, *C. parapsilosis*, *C. glabrata*, *C.dubliniensis*	Planktonic/biofilm	MIC	0.1–0.5 mg/mL	-	[65]
N/R	*C. albicans*, *S. aureus*	Planktonic Biofilm	MIC/Biofilm percentag	200 µg/mL	-	[66]
N/R	*C. albicans*	Biofilm	aPDT	1.5 g/mL	20.1 J/cm^2^	[67]
DMSO (10%)	*C. albicans*	Biofilm	aPDT	20, 40, 60 and 80 µM	2.64, 5.28, 7.92, 10.56, and 13.2 J/cm^2^	[68]
DMSO (1%)	*C. albicans*	Biofilm	aPDT	40 and 80 mM	37.5 and 50 J/cm^2^	[69]
N/R	*C. albicans*	Biofilm	aPDT	100 µM	10 J/cm^2^	[70]
DMSO (2.5%)	Fluconazole-resistant *C. albicans*	Planktonic/biofilm/infection model	MIC/ aPDT	40 µM	5.28 J/cm^2^	[71]
	Fluconazole-susceptible *C. albicans*			80 µM	40.3 J/cm^2^	
DMSO	*C. albicans*, *F. oxysporum*, *A. flavus*, *A. niger*, *C. neoformans*	Planktonic	MIC	137.5–200 μg/mL	-	[72]

-: not performed. N/R: not reported. N/C: not clear.

**Table 2 ijms-22-07130-t002:** Antimicrobial studies performed with CUR in micelles.

Type of Micelles	[CUR] Formulation	Microorganism	Type of Culture	Antimicrobial Method	Antimicrobial [CUR]	Light/Ultrasonic Parameters	Reference
Mixed polymer micelles	1000 ppm	*E. coli*, *S. aureus*, *A. niger*	Planktonic	MIC	350 and 275 µg/mL	-	[92]
PCL-*b*-PAsp and Ag	2 mg/mL	*P. aeruginosa*, *S. aureus*	Planktonic	OD_600nm_	8–500 µg/mL	-	[93]
mPEG-OA	1:10	*P. aeruginosa*	Planktonic	MIC	400 µg/mL	-	[94]
PEG-PCL	10 mg	*C. albicans*	Planktonic	MIC	256 µg/mL	-	[95]
PEG-PE	50 mM	*S. mutans*	Planktonic	SACT	50 mM	1.56 W/cm^2^	[96]
DAPMA, SPD, SPM	0.32 mg/mL	*P. aeruginosa*	Planktonic	OD_600nm_ and aPDT	250, 500 nM, 1 µM and 50, 100 nM	18 and 30 J/cm^2^	[97]
P123	0.5% *w*/*V*	*S. aureus*	Planktonic	aPDT	7.80 μmol/L	6.5 J/cm^2^	[98]
PCL-*b*-PHMG-*b*-PCL, STES	10 mg	*S. aureus*, *E. coli*	Planktonic	MIC	16 and 32 μg/mL *	-	[99]
CUR-PLGA-DEX	1 mg/mL	*P. fluorescens*, *P. putida*	Planktonic biofilm	OD_600nm_ antibiofilm	0.625–5 mg/mL	-	[100]

[CUR]: CUR concentration. -: not performed. *: MIC of micelle without CUR.

**Table 3 ijms-22-07130-t003:** Antimicrobial studies performed with CUR in liposomes and solid lipid nanoparticles (SLN).

Type of Liposomes or SLN	[CUR] Formulation	Microorganism	Type of Culture	Antimicrobial Method	Antimicrobial [CUR]	Reference
Lecithin and cholesterol	0.5 mg/mL	*A. sobria*, *C. violaceum*, *A. tumefaciens*	Planktonic biofilm	MIC, antibiofilm	420, 400, and 460 μg/mL	[105]
PCNL	60.65 ± 1.68 µg/µL	*B. subtilis*, *K. pneumoniae*, *C. violaceum*, *E. coli*, *M. smegmatis*, *A. niger*, *C. albicans*, *F. oxysporum*	Planktonic	Disk diffusion assay	N/R	[106]
Phosphocolines	100:1 M	*S. aureus*	Planktonic	MIC	7 μg/mL	[107]
PLGA: triglycerides: F68	0.8 mg/mL	*E. coli*, *S. typhimurium*, *P. aeruginosa*, *S. aureus*, *B. sonorensis*, *B. licheniformis*	Planktonic	MIC	75 and 100 μg/mL	[108]
Soya lecithin and menthol	0.5 mg/mL	MRSA	Planktonic Biofilm	MIC, microscopy, biomass	10 and 125 µg/mL	[109]
CurSLN	60 mg/500 mg lipid	*S. aureus*, *S. mutans*, *V. streptococci*, *L. acidophilus*, *E. coli*, *C. albicans*	Planktonic	MIC MBC	0.09375–3 and 1.5–6 mg/mL	[110]

[CUR]: CUR concentration. N/R: not reported.

**Table 4 ijms-22-07130-t004:** Antimicrobial studies performed with curcumin/curcuminoid in emulsions.

Type of Emulsion	[CUR] Formulation	Microorganism	Type of Culture	Antimicrobial Method	Antimicrobial Concentration	Light/Ultrasonic Parameters	Reference
THC ME	5%	HIV-1	Cell infection	IC50	0.9357 μM	-	[116]
CUR-NE	N/R	HPV	-	aPDT	80 µM	50 J/cm^2^	[117]
CUR-NE	N/R	DENV-1 to 4	Cell infection	Cell viability	1, 5, 10 µg/mL	-	[118]
P60-CUR	4 mg/L	E. coli	Planktonic	OD 595 nm	N/R	-	[119]
PE:CUR	0.566 mg/mL	*S. aureus*, *S. epidermidis*, *S. faecalis*, *C. albicans*, *E. coli*	Planktonic	Inhibition zone	1 mg/mL *	-	[120]
cu-SEDDS	1%	*S. aureus*, *E. coli*, *P. aeruginosa*, *K. pneumonia*	Planktonic	MIC	45–62 µg/mL	-	[121]
CUR:NE in microbeads	0.5 mg/mL	*E. coli*, *S. typhmerium*, *Y. enterocolitica*, *P. aeruginosa*, *S. aureus*, *B. cereus*, *L. monocytogenes*	Planktonic	Inhibition zone	90 and 180 mg/mL *	-	[122]
Lignin sulfomethylated	0.3 mg/mL	*S. aureus*	Planktonic	OD 600 nm	2.4 mg/mL *	-	[123]
C14-EDA/GM/W C14-MEDA/GM/W	N/R	*C. albicans*	Planktonic, biofilm	Microdilution assay, antibiofilm	100 µg/mL, 20 µg	-	[124]

[CUR]: CUR concentration. -: not performed. N/R: not reported. *: formulation concentration.

**Table 5 ijms-22-07130-t005:** Antimicrobial studies performed with CUR in CDs.

Type of CD	[CUR] Formulation	Microorganism	Type of Culture	Antimicrobial Method	Antimicrobial [CUR]	Light/Ultrasonic Parameters	Reference
PEG-based β-CD or γ-CD	10 µM	*E. coli*, *E. faecalis*	Planktonic	aPDT	10 µM	4.8, 29 J/cm^2^	[133]
HPMC-stabilized hydroxypro pyl-β-CD	7.64 × 10^−3^ M	*E. coli*	Planktonic	aPDT	10, 25 µM	5, 14, 28 J/cm^2^	[134]
methyl-β-CD hyaluronic acid HPMC	7.64 × 10^−3 ^M	*E. faecalis*, *E. coli*	Planktonic	aPDT	0.5–25 µM	11, 16, 32 J/cm^2^	[135]
carboxymethyl-β-CD	20 µM	*E. coli*	Planktonic	aPDT	0.7 ± 0.1 to 4.1 ± 1.6 nmole cm^−2^	1050 ± 250 lx	[136]
hydrogel with CUR in hydroxypropyl-β-CD	15.8 mg/mL	*S. aureus*	Planktonic	Inhibition zone	2% (*w*/*v*)	-	[137]
α- and β-CD	1 mol/L	*E. coli*, *S. aureus*	Planktonic	MIC, OD 600 nm	0.25 and 0.31 mg/mL	-	[138]
β-CD or γ-CD in CS	0.06 mM	*E. coli*, *S. aureus*	Planktonic	MIC, Zone of inhibition	64 and 32 µg/mL	-	[139]
γ-CD	25 mg/L	*T. rubrum*	Planktonic	MIC, aPDT	N/R	45 J/cm^2^	[140]
hydroxypropyl-β-CD	1:1	*B. subtillis*, *S. aureus*, *S. pyrogenes*, *P. aeruginosa*, *C. difficile*, *C. butyricum*, *L. monocytogenes*, *E. faecalis*, *E. coli*, *K. pneumoniae*, *P. mirabilis*, *S. typhimurium*, *E. aerogens*, *C. kusei*, *C. albicans*	Planktonic	Inhibition zone	25 mg/mL	-	[141]
methyl-β-CD	20 mM	*E. coli*	Planktonic	MIC, MBC, aPDT	500, 90 µM	9 J/cm^2^	[142]

[CUR]: CUR concentration. -: not performed. N/R: not reported.

**Table 6 ijms-22-07130-t006:** Antimicrobial studies performed with CUR in CS.

Type of CS	[CUR] Formulation	Microorganism	Type of Culture	Antimicrobial Method	Antimicrobial [CUR]	Reference
PEG-CS	4.4%, 5 mg/mL	MRSA, *P. aeruginosa*	Planktonic, Animal model	OD_600nm_, CFU	5 and 10 mg/mL *	[148]
CCS microspheres	12.27 mg/mL, 1 mol	*S. aureus*, *E. coli*	Planktonic	Zone of inhibition, MIC	N/R	[149]
CS nanoparticles	1.06 mg/mL	*S. mutans*	Planktonic, biofilm	MIC	0.114 mg/mL	[150]
CS-CMS-MMT	0.0004–0.004 g	*S. mutans*	Planktonic, Biofilm	MIC	0.101 mg/mL	[151]
CS-GP-CUR	148.09 ± 5.01 µg	*S. aureus*	Planktonic	Zone of inhibition, tissue bacteria count	N/C	[152]
PVA-CS-CUR	N/C	*E. coli*, *P. aeruginosa*, *S. aureus*, *B. subtilis*	Planktonic	Zone of inhibition	N/R	[153]
PVA-CS-CUR	10, 20, 30 mg	*P. multocida*, *S. aureus*, *E. coli*, *B. subtilis*	Planktonic	Zone of inhibition	10, 20, 30 mg	[154]
CS NPs	2, 4, 8, 16%	*C. albicans*, *S. aureus*	Planktonic, Biofilm	MIC, Colony count	400 mg/mL	[155]
CS NPs	4 mg/mL	HCV-4	N/R	Antiviral assay	15 µg/mL	[156]
CS/milk protein nanocomposites	100 mg	PVY	Plant infection	Antiviral activity	500, 1000, 1500 mg/100 mL	[157]

[CUR]: CUR concentration. N/R: not reported. N/C: not clear. *: formulation concentration.

**Table 7 ijms-22-07130-t007:** Antimicrobial studies performed with curcumin in polymeric drug delivery systems.

Type of Polymeric DDS	[CUR] Formulation	Microorganism	Type of Culture	Antimicrobial Method	Antimicrobial [CUR]	Light/Ultrasonic Parameters	Reference
PEG 400 γ-CD and PEG + β-CD	0.18%	*E. faecalis* *E. coli*	Planktonic	CFU/mL aPDT	N/R	9.7 J/cm^2^ 29 J/cm^2^	[158]
CUR-NP without polymer	100 mg	*S. aureus* *B. subtillis* *E. coli* *P. aeruginosa* *P. notatum* *A. niger*	Planktonic	MIC Inhibition zone	100 mg 0.27 mmol	-	[159]
CUR-NP without polymer	100 mg	*M. lutues* *S. aureus* *E. coli* *P. aeruginosa*	Planktonic	MBC	N/R	-	[160]
Mixed polymer NP	5 mM	E. coli	Planktonic	MIC	400–500 μM	-	[161]
CTAB Tween 20 Sodium dodecylsulfate	100 mg/mL	*L. monicytogenes*	Planktonic	Inhibition zone	N/R	-	[162]
PLA/dextran sulfate	4 mg/mL	MRSA *C. albicans* *S. mutans*	Planktonic/mono- and –mixed biofilm	aPDT	260 μM	43.2 J/cm^2^	[163]
PLA/dextran sulfate	0.4%	*C. albicans*	Animal model	aPDT	260 μM	37.5 J/cm^2^	[164]
Nanocurcumin	N/R	*P. aeruginosa* (isolates) and standard strain	Planktonic	MIC	128 µg/mL	-	[165]
PLGA	5 mg	*S. saprophyticus* subsp. *Bovis* *E. coli*	Planktonic	aPDT	50 µg/mL	13.2 J/cm^2^	[166]
Eudragit L-100	N/C	*L. monocytogenes*	Planktonic	Animal model infection	N/R	-	[167]
nCUR	N/R	*S. mutans*	Planktonic Biofilm	Inhibition zone aPDT	N/R	300–420 J/cm^2^	[168]
nCUR combined with indocyanine	100 mg	*E. faecalis*	Biofilm	Metabolic activity	N/R	500 mW/cm^2^	[169]
PVAc-CUR-PET-PVDC	0.02 g	*S. aureus* *S. tiphimurium*	Planktonic	aPDT	N/R	24, 48, and 72 J/cm^2^	[170]
MOA.CUR-PLGA-NP	Up to 10%	S. mutans	Biofilm	aPDT	7% wt	45 J/cm^2^	[171]
CS- β-CD	N/C	*S. aureus* *E. coli*	Planktonic	Colony count	Up to 0.03%	-	[172]

[CUR]: CUR concentration. -: not performed. N/R: not reported. N/C: not clear.

**Table 8 ijms-22-07130-t008:** Antimicrobial studies performed with CUR complexes with metallic NPs.

Type of Metallic Material	[CUR] Formulation	Microorganism	Type of Culture	Antimicrobial Method	Antimicrobial [CUR]	Reference
CUR-AgNPs	20 mg/mL	*P. aeruginosa*, *E. coli*, *B. subtilis*, *S. aureus*	Planktonic	MIC	20 mg/mL	[177]
Ag-CUR-nanoconjugates	0.1 mM	*E. coli*, *Salmonella* spp., *Fusarium* spp., *S. aureus*	Planktonic	Zone of Inhibition	0.1 mM	[178]
AgCURNPs	500 mg	*P. aeruginosa* *S. aureus*	Biofilm	CLSM SEM	Up to 400 μg/mL	[179]
AgNPs	7 mg	*E. coli*	Planktonic	Turbidimetric Assay	0.005 µM	[180]
cAgNPs	7 mg	*E. coli* *B. subtilis*	Planktonic	MIC CFU/mL	7 mg	[181]
Ru II complex	0.092 g	*E. coli*, *S. aureus*, *K. pneumoniae*, *A. baumannii*,. *P. aeruginosa*, *Enterococcus* sp.	Plakntonic	MIC/FICI	>64 µg/mL	[182]
SCMC SNCF nanocomposites with CUR	0.25 mg/mL	*E. coli*	Planktonic	Disc Method Count Method	2 mg/mL	[183]
CSCL CUR-AgNP	0.092 g	*E. coli*, *B. subtilis*	Planktonic	Zone of Inhibition	10 and 20 μM	[184]
nSnH	10%	*S. aureus* *E. coli*.	Planktonic	CFU/mL	N/R	[185]
Nanocomposite of CUR and ZnO NPs	N/C	*S. epidermidis* *S. hemolyticus* *S. saprophyticus*	Planktonic	Zone of Inhibition	1000, 750, 500, 250 μg/mL	[186]
Thermo-responsive hydrogels	N/C	*S. aureus* *P. aeruginosa* *E. coli*	Planktonic	MIC	400 μg/mL	[187]
CUR-AgNPs	5 mg/mL	*C. albicans*, *C. glabrata*, *C. tropicalis*, *C. parapsilosis*, *C. krusei*, *C. kefyr*	Planktonic	Zone of Inhibition, MIC	32.2–250 μg/mL	[188]
Gel-CUR-Ag	20 mg	*P. aeruginosa* *S. aureus*	Planktonic	MIC MBC	20 mg	[189]
HGZ-CUR	N/C	*S. aureus* *T. rubrum*	Planktonic	Zone of Inhibition	N/C	[190]
CHG-ZnO-CUR	N/C	*S. aureus* *T. rubrum*	Planktonic	Zone of Inhibition	N/C	[191]
Copper (II) oxide NPs	1 g	*E. faecalis* *P. aeruginosa*	Planktonic	Zone of Inhibition CFU/mL	1 mg/mL	[192]
OA-Ag-C	1 g	*P. aeruginosa* *S. aureus*	Planktonic	OD_600nm_	2.5 mg/mL	[193]
Ag-NP-β-CD-BC	0.79 g	*P. aeruginosa*, *S. aureus*, *C. auris*	Planktonic	Zone of Inhibition	N/R	[194]
Cotton fabrics coated ZnO-NP	2.71 × 10^−3^ M	*S. aureus*, *E. coli*	Planktonic	Bacterial Count	N/R	[195]
CS-ZnO-CUR	0.2 g	*S. aureus*, *E. coli*	Planktonic	MIC MBC	Up to 50 μg/mL	[196]
CUR-TiO_2_ -CS	100–300 mg	*S. aureus*, *E. coli*	Planktonic Animal infection	MIC	10 mg	[197]
CUR-Au-NPs	1 mg/mL	*E. coli*, *B. subtilis*, *S. aureus*, *P. aeruginosa*	Planktonic	Zone of Inhibition	100, 200, 300 μg/mL	[198]

[CUR]: CUR concentration. N/R: not reported. N/C: not clear.

**Table 9 ijms-22-07130-t009:** Antimicrobial studies performed with CUR in porous DDSs.

Porous DDS	[CUR] Formulation	Microorganism	Type of Culture	Antimicrobial Method	Antimicrobial [CUR]	Light/Ultrasonic Parameters	Reference
Cu-SNP/Ag	1.0 mmol	*E. coli*	Planktonic	aPDT	N/R	72 J/cm^2^	[204]
Bionanocomposite silica/chitosan	100 mg	*E. coli* *S*. *aureus*	Planktonic	Zone of inhibition	N/R	-	[205]
NCIP	1 mg	HIV-1	Transfected cells	Immunofluorescent staining	5–8 mg/mL	-	[206]
Lollipop-like MSN	30 mg L^−1^	*E. coli* *S. aureus*	Planktonic	OD_600nm_	N/R	-	[207]
SBA-15/PDA/Ag	2 mg	*E. coli* *S. aureus*	Planktonic	CFU/mL	50 mM	-	[208]

[CUR]: CUR concentration. -: not performed. N/R: not reported.

**Table 10 ijms-22-07130-t010:** Antimicrobial studies performed with CUR in the composite of graphene and quantum dots (QDs).

Type of Material	[CUR] Formulation	Microorganism	Type of Culture	Antimicrobial Method	Antimicrobial [CUR]	Light/Ultrasonic Parameters	Reference
**G-NH_2_–IONP–PEG**	0.004 g	*S. aureus* *E. coli*	Planktonic	Colony count	100, 125 μg/mL	-	[210]
**CUR-rGO**	N/R	*E. faecalis*	Biofilm	MBIC aPDT	250 μg/mL	360 J/cm^2^	[211]
**GrZnO**	N/R	MRSA	Planktonic	MIC Inhibition zone Metabolic activity	Up to 62.5 μg/mL	-	[212]
**CUR-cQDs**	0.6	*S. aureus* MRSA *E. faecalis* *E. coli* *K. pneumoniae* *P. aeruginosa*	Planktonic Biofilm	Grown inhibition Biomass evaluation Confocal microscopy	3.91–7.825 µg/mL	-	[213]
**CUR-cQDs**	200 mg	EV-71	Cell infection Animal infection	MIC Plaque assay TCIC_50_ assay Western blot PCR	5 μg/mL	-	[214]
**CUR-MQD**	2:1 wt%	*K. pneumoniae* *P. aeruginosa* *S. aureus*	Planktonic	MIC MBC Confocal microscopy Fluorescence microscopy Flow cytometry	<0.00625–0.125 μg/mL	-	[215]
**CUR-GQDs**	N/C	*A. actinomycetemcomitans* *P. gingivalis* *P. intermedia*	Mixed- biofilm	aPDT	100 μg/mL	60–80 J/cm^2^	[216]

[CUR]: CUR concentration. -: not performed. N/R: not reported. N/C: not clear.

**Table 11 ijms-22-07130-t011:** Antimicrobial studies performed with CUR in films, hydrogels, and other nanomaterials.

Type of Material	[CUR] Formulation	Microorganism	Type of Culture	Antimicrobial Method	Antimicrobial [CUR]	Light/Ultrasonic Parameters	Reference
CuR-SiNPs	20 mg	*S. aureus* *P. aeruginosa*	Planktonic Biofilm	aPDT	50 μg/mL 1 mg/mL	20 J/cm^2^	[218]
CUR-HNT-DX	10 mg	*S. marcescens* *E. coli*	Planktonic Infection model	Grown inhibition, Confocal microscopy	Up to 0.5 mg/mL	-	[219]
Exosomes	N/R	HIV-1 infection	-	Flow cytometry	N/R	-	[220]
Electrospun nanofibers	100 mg/mL	*Actinomyces naeslundii*	Biofilm	aPDT	2.5 and 5 mg/mL	1200 mW/cm^2^	[221]
Ga NF CD-GO NF	0.1 mol	*B. cereus* *E. coli*	Planktonic	Zone of inhibition MIC	Up to 63.25 µg/mL	-	[222]
Multinanofibers-film	1, 2.5, and 5 mg/mL	*S. aureus* *E. coli*	Planktonic	UFC/mL Confocal microscopy	1 mg/mL	-	[223]
Nanofibers scaffolds	4.0 wt%	*S. aureus* *Pseudomonas* sp.	Planktonic	Colony count	N/R	-	[224]
Nanofibrous scaffold	5%	*S. aureus* *E. coli*	Planktonic	Colony count	20 mg	-	[225]
Nanofibers	5 and 10%wt	*S. aureus* *E. coli*	Planktonic	OD_600nm_	Up to 212.5 µg/mL	-	[226]
CSDG	1 w/w	*S. aureus* *E. coli*	Planktonic Infection model	Colony count Microscopy	N/R	-	[227]
Gelatin film	0, 0.25, 0.5, 1.0, and 1.5 wt%	*E. coli* *L. monocytogenes*	Planktonic	UFC/mL	0.25 and 1.5 wt%	-	[228]
ZnO-CMC film	0.5 and 1.0 wt%	*E. coli* *L. monocytogenes*	Planktonic	UFC/mL	1 wt%	-	[229]
Pectin film	40 mg	*E. coli* *L. monocytogenes*	Planktonic	UFC/mL	N/R	-	[230]
Edible film	0.4% (*w*/*v*)	*E. coli* *B. subtilis*	Planktonic	Zone of inhibition	1% wt.	-	[231]

[CUR]: CUR concentration. -: not performed. N/R: not reported. N/C: not clear.

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
