# Peer review of "Antimicrobial Activity of Curcumin in Nanoformulations: A Comprehensive Review"

_ijms, 2021, doi:10.3390/ijms22137130_

Round 1

Reviewer 1 Report

GENERAL CONSIDERATIONS

  1. Considering that this is a review work, some recent papers should be taken into account. The following table resume for each section the papers should be mentioned:

Section 3.1

Barros CHN, Hiebner DW, Fulaz S, Vitale S, Quinn L, Casey E. Synthesis and self-assembly of curcumin-modified amphiphilic polymeric micelles with antibacterial activity.                                                           

 J Nanobiotechnology. 2021 Apr 13;19(1):104. doi: 10.1186/s12951-021-00851-2. PMID: 33849570; PMCID: PMC8045376.

Section 3.2

Ding T, Li T, Wang Z, Li J. Curcumin liposomes interfere with quorum sensing system of Aeromonas sobria and in silico analysis.                                                   

 Sci Rep. 2017 Aug 17;7(1):8612. doi: 10.1038/s41598-017-08986-9. PMID: 28819178; PMCID: PMC5561023.

 Section 3.6

N. Muniyappan, M. Pandeeswaran, Augustine Amalraj,

Green synthesis of gold nanoparticles using Curcuma pseudomontana isolated curcumin: Its characterization, antimicrobial, antioxidant and anti- inflammatory activities,

Environmental Chemistry and Ecotoxicology,

Volume 3,

2021,

Pages 117-124,

ISSN 2590-1826,

https://doi.org/10.1016/j.enceco.2021.01.002.

Loan Khanh, L., Thanh Truc, N., Tan Dat, N., Thi Phuong Nghi, N., van Toi, V., Thi Thu Hoai, N., Ngoc Quyen, T., Thi Thanh Loan, T., & Thi Hiep, N. (2019). Gelatin-stabilized composites of silver nanoparticles and curcumin: characterization, antibacterial and antioxidant study.                                                                           

Science and technology of advanced materials, 20(1), 276–290. https://doi.org/10.1080/14686996.2019.1585131

Paul S, Mohanram K, Kannan I. Antifungal activity of curcumin-silver nanoparticles against fluconazole-resistant clinical isolates of Candida species. Ayu. 2018 Jul-Sep;39(3):182-186. doi: 10.4103/ayu.AYU_24_18. PMID: 31000996; PMCID: PMC6454909.

  1. The format of each table should be changed so as to have one column describing the Curcumin concentration in the formulation and another column representing the MIC. For example, in table n. 3 the row relatives to  Mittal et al., 2019 describes the concentration of Curcumin in the formulation while the row corresponding to  Gao et al., 2020 represents the MIC concentration

MINOR REVISIONS

Line 292

“Antimicrobials” should be replaced with “antimicrobial”.

Line 375-376

The wavelength mentioned in the original research paper [Ref. 97] is 445nm not 455 nm.

TABLE 3

Gao et al., 2020

Up to 100 µg/mL?? ; CUR at a concentration of 0.8 mg ml−1

Bathia et al., 2021?? The name of the ref. Is not correct. It should be Bhatia et al., 2021.  

TABLE N.4

Ngwabebhoh et al., 2018

E. coli is written two times.

Khan et al.,2019

cu-SEDDS not SEDDS.

Chen et al., 2019

Type of emulsion HIPEs-cur.  Curcumin conc. 30 g?? Could you please explain??

Section 3.4 was mentioned two times mistakenly. (3.4 CUR in Cyclodextrin and 3.4 CUR in Chitosan)

Author Response

            Thank you for the considerations and careful revision of our manuscript. All changes done in the manuscript are highlighted in red. A private editing service company performed the English revision of the manuscript (the certificate is attached at the end of this document).

  1. “Considering that this is a review work, some recent papers should be taken into account. The following table resume for each section the papers should be mentioned:”

The five papers indicated were included in the manuscript. One of them was already in the manuscript (Loan Khanh et al. 2019, reference 186), although the authors’ names were not properly cited. This correction was also done.

  1. “The format of each table should be changed so as to have one column describing the Curcumin concentration in the formulation and another column representing the MIC. For example, in table n. 3 the row relatives to Mittal et al., 2019 describes the concentration of Curcumin in the formulation while the row corresponding to Gao et al., 2020 represents the MIC concentration”

            Thank you so much for such observation. All Tables summarizing the drug delivery systems for curcumin (Tables 2 to 11) were changed and now they have one column with the curcumin concentration used in the formulation and another column with the effective antimicrobial concentration of the nanocurcumin.

“MINOR REVISIONS

Line 292          “Antimicrobials” should be replaced with “antimicrobial”.”

            The term “antimicrobials” were replaced with “antimicrobial”.

“Line 375-376            The wavelength mentioned in the original research paper [Ref. 97] is 445nm not 455 nm.”

            The wavelength was corrected: 445 nm.

“TABLE 3       Gao et al., 2020          Up to 100 µg/mL?? ; CUR at a concentration of 0.8 mg ml−1

            The concentration up to 100 µg/mL is the MIC of curcumin against the microbial species evaluated in the paper. The concentration of 0.8 mg/mL is shown in Figure 1 of the original paper and is the concentration of curcumin in its free form and in nanocapsules. Both concentrations were now included in Table 3.

“Bathia et al., 2021?? The name of the ref. Is not correct. It should be Bhatia et al., 2021.”

            The correction was done: Bhatia et al., 2021.

“TABLE N.4   Ngwabebhoh et al., 2018        E. coli is written two times.”

            The correction was done: E. coli appears once now.

“Khan et al.,2019       cu-SEDDS not SEDDS.”

            The correction was done: cu-SEDDS.

“Chen et al., 2019       Type of emulsion HIPEs-cur. Curcumin conc. 30 g?? Could you please explain??”

            The correct MIC of curcumin in this study is 2.4 mg/mL. The correction was done.

“Section 3.4 was mentioned two times mistakenly. (3.4 CUR in Cyclodextrin and 3.4 CUR in Chitosan)”

            The correction was done. Additionally, a section about CUR in Solid Lipid Nanoparticles was added separately from CUR in liposomes (they were together in the original version of the manuscript). The subsequent numbers (3.4 to 3.12) were also changed.

Reviewer 2 Report

This article is interesting. This is a huge review of antimicrobial activity of curcumin in nanoformulations. In this manuscript there are big tables. I would suggest reducing them and making more user-friendly and readable. As conclusion, I suggest to write a brief note on the future of curcumin use.

Author Response

“This article is interesting. This is a huge review of antimicrobial activity of curcumin in nanoformulations. In this manuscript there are big tables. I would suggest reducing them and making more user-friendly and readable. As conclusion, I suggest to write a brief note on the future of curcumin use.”

            Thank you so much for the revision and the comments that improved our manuscript. All changes done in the manuscript are highlighted in blue. We agree that the manuscript has big tables, especially table 1 (free curcumin). However, our idea was to mention the effective antimicrobial concentration of curcumin from each study reviewed in our paper. In our opinion, showing the effective concentration in tables was more appropriate than describing them in the text, mainly in the section of free curcumin where there are several studies reported and this section would be longer. Therefore, reporting the effective concentration in the table would summarize better this important information for the reader than describing it in the text. In addition, reviewer 1 asked for including more studies of curcumin in nanoformulations in our manuscript. Thus, we thought that citing each study in the tables would summarize properly the selected papers.

            A brief note on the future of curcumin use was written as suggested. We believed that curcumin in drug delivery systems and nanoformulations will not replace conventional antimicrobials, but play an adjuvant role in the treatment of infections. Additionally, the stimuli-responsive nanocarriers are promising systems to improve the efficacy and the drug delivery at the target tissue.

Round 2

Reviewer 1 Report

The authors made the required correction therefore the review is now publishable.